# Gain control of sensory input across polysynaptic circuitries in mouse visual cortex by a single G protein-coupled receptor type (5-HT$_{2A}$)

Ruxandra Barzan[1,2,6], Beyza Bozkurt [1,2], Mohammadreza M. Nejad[3], Sandra T. Süß [4], Tatjana Surdin [4], Hanna Böke[4], Katharina Spoida [4], Zohre Azimi[1,2], Michelle Grömmke[5], Dennis Eickelbeck[4], Melanie D. Mark [5], Lennard Rohr[4], Ida Siveke[4], Sen Cheng [3], Stefan Herlitze[4] & Dirk Jancke [1,2] ✉

Response gain is a crucial means by which modulatory systems control the impact of sensory input. In the visual cortex, the serotonergic 5-HT$_{2A}$ receptor is key in such modulation. However, due to its expression across different cell types and lack of methods that allow for specific activation, the underlying network mechanisms remain unsolved. Here we optogenetically activate endogenous G protein-coupled receptor (GPCR) signaling of a single receptor subtype in distinct *mouse* neocortical subpopulations in vivo. We show that photoactivation of the 5-HT$_{2A}$ receptor pathway in pyramidal neurons enhances firing of both excitatory neurons and interneurons, whereas 5-HT$_{2A}$ photoactivation in parvalbumin interneurons produces bidirectional effects. Combined photoactivation in both cell types and cortical network modelling demonstrates a conductance-driven polysynaptic mechanism that controls the gain of visual input without affecting ongoing baseline levels. Our study opens avenues to explore GPCRs neuromodulation and its impact on sensory-driven activity and ongoing neuronal dynamics.

Neuromodulation of cortical processing has a substantial effect on neural response gain, which allows adaptive and flexible perception, cognition, and behavior[1–3]. However, the interactions across neuronal circuits underlying gain control remain generally puzzling and intensely debated[4–7], as the results and conclusions heavily depend on experimental conditions and the configuration of the network in which neurons are embedded[8]. Another structural characteristic of brain modulatory systems is the co-distribution of their receptor families across various types of neurons[9–13]. This variety in expression patterns[14] additionally hinders experimental and theoretical access to

their coherent modulatory network function in vivo. In particular, receptors activated by dopamine, serotonin, acetylcholine, or noradrenaline can have opposing downstream effects on target neurons via their respective G protein-coupled receptors (GPCRs). Hence, the individual contribution of a single receptor type to the net functional output of a given cortical area remains largely obscure[9,15–19].

Here, we exploited in vivo optogenetic control of the pathway of a single GPCR subtype, 5-HT$_{2A}$, specifically activated in pyramidal and parvalbumin (PV) neurons in the *mouse* primary visual cortex (V1). Importantly, the 5-HT$_{2A}$ C-terminus of the optogenetic tool used

[1]Optical Imaging Group, Institut für Neuroinformatik, Ruhr University Bochum, Bochum, Germany. [2]International Graduate School of Neuroscience, Ruhr University Bochum, Bochum, Germany. [3]Computational Neuroscience, Institute for Neural Computation, Ruhr University Bochum, Bochum, Germany. [4]Department of Zoology and Neurobiology, Ruhr University Bochum, Bochum, Germany. [5]Behavioral Neuroscience, Ruhr University Bochum, Bochum, Germany. [6]Present address: MEDICE Arzneimittel Pütter GmbH & Co. KG, Iserlohn, Germany. ✉e-mail: dirk.jancke@rub.de

enables its expression in the endogenous receptor-specific cellular domains[20], providing a light-activatable functional equivalent to endogenous 5-HT$_{2A}$ receptor signals[20–25]. Moreover, targeting GPCR tools to receptor-specific domains triggers downstream kinetics with similar strength and ensures no overshoot compared to their native second messenger pathway effects[25].

An increase in the cortical serotonin (5-HT) levels by micro-iontophoresis in V1 altered the gain of visual responses in rodents[26] and in non-human primates[27,28]. Further pharmacological studies manipulating 5-HT receptor activation revealed a dominant contribution of the 5-HT$_{2A}$ receptor to gain modulation[27,29–31] (Fig. 1a). However, these findings raise fundamental questions about the circuit mechanisms underlying such gain control. First, the 5-HT$_{2A}$ receptor pathway is excitatory in single cells[32], leaving unanswered how the in vivo net-

work generates an overall suppression upon its activation. Second, suppression of visual sensory activity was found to scale divisively with little to no influence on baseline[20,29–31]. Again, the circuit mechanisms reducing the magnitude gain of sensory responses without affecting spontaneous ongoing activity levels (Fig. 1b) remain to be identified. Hence, in this study, we isolate a specific coherent function of a modulatory system, such as the divisive gain modulation[33,34] of sensory input by the 5-HT system, and demonstrate the network mechanisms by which this function is shaped across different types of neurons.

## Results

We devised a method for optogenetic control of a single endogenous receptor pathway (5-HT$_{2A}$) to explore basic polysynaptic mechanisms of GPCR signaling in two different types of cortical neurons (i.e., pyr-

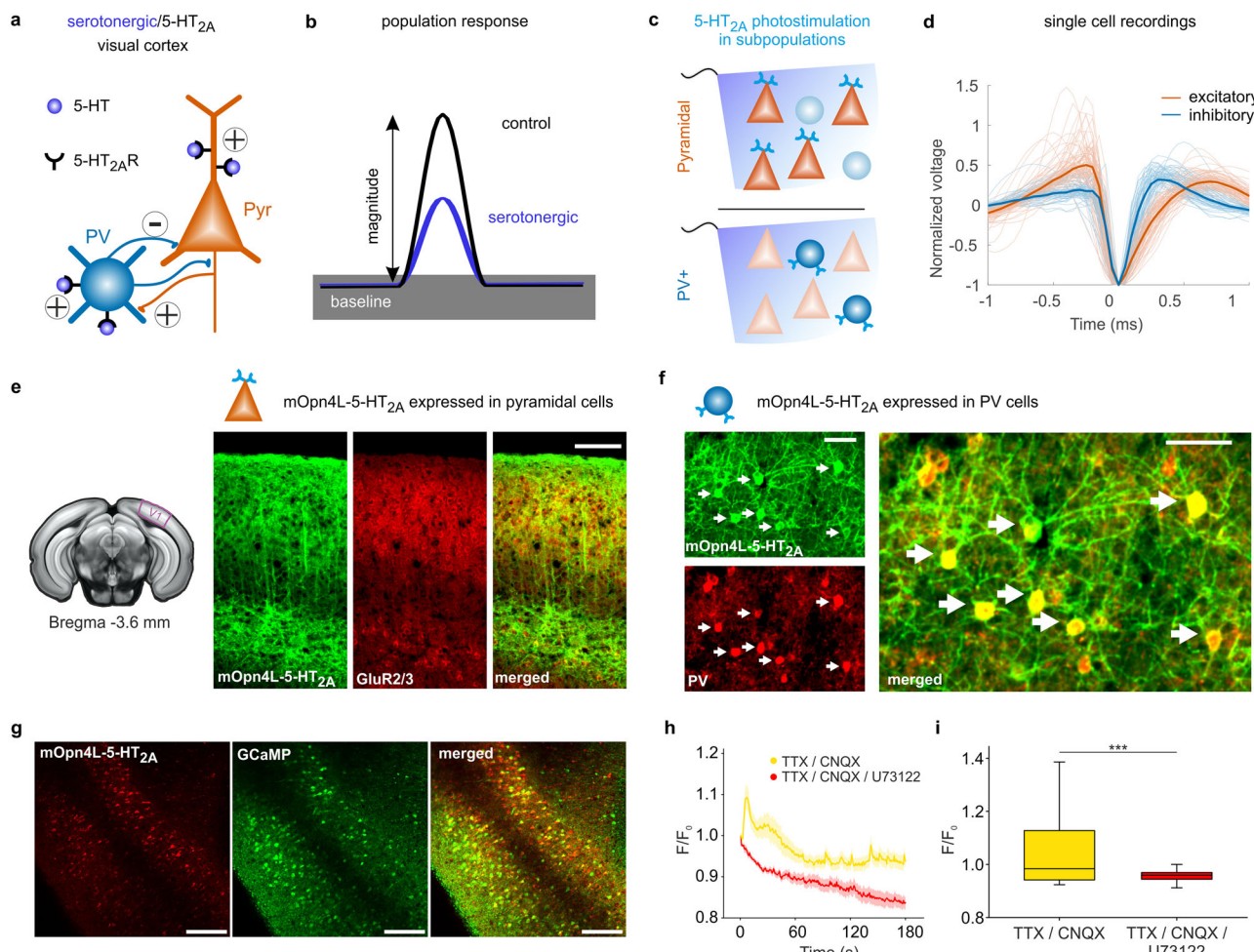

**Fig. 1 | Gain control in the visual cortex via serotonergic receptors. a** Schematics representing the localization and main modulatory synaptic effects of 5-HT$_{2A}$ receptors expressed in pyramidal and parvalbumin (PV) neurons in the *mouse* cortex. **b** 5-HT$_{2A}$ receptors are known to reduce the gain of visually evoked responses without affecting baseline levels of activity. **c** 5-HT$_{2A}$ receptor activation in populations of either pyramidal or PV neurons was controlled optogenetically by light. **d** Silicon probe recordings allowed source separation of responses of putative excitatory or inhibitory neurons based on analysis of waveform features (see "Methods" section). **e** Left to right: coronal slice of the *mouse* brain with V1 location marked; confocal scans of slices with mOpn4L-5-HT$_{2A}$ expression (green) in a NEX-Cre *mouse*, antibody against GluR2/3 (red) and the merged image. **f** Same as (**e**) for a PV-Cre *mouse* with antibody against PV (red). Arrows point to double-positive cells. The scans in (**e**, **f**) represent areas enlarged in Extended Data Fig. 3b and Extended

Data Fig. 4a, respectively. **g** 2-Photon fluorescence images of V1 cortical slice from a NEX-Cre *mouse* showing expressing of mOpn4L-mCherry-5-HT$_{2A}$ (left) and GCaMP (middle) in pyramidal neurons, merged image (right). Images in (**e**–**g**) are representative of three independent experiments. **h** Time course of Ca$^{2+}$-dependent changes in fluorescence during 3 min blue light activation under the influence of TTX/CNQX ($n = 103$ cells) or TTX/CNQX/U73122 (see "Methods" section) ($n = 50$ cells). Traces and shadings represent mean ± SEM. **i** Comparison of the amplitude of all cells depicted in (**h**) at the time of peak during TTX/CNQX and TTX/CNQX/ U73122 applications. Box plots indicate median (middle line), 25th, 75th percentile (box), 10th, and 90th percentile (whiskers), ***$p = 0.0007$, two-sided Mann–Whitney U-test. Scale bars: 100 μm in (**e**), 50 μm in (**f**), and 25 μm in (**g**). Source data are provided as a Source Data file.

amidal and PV neurons in the visual cortex; see Fig. 1c–f). To achieve this, we expressed a chimeric construct consisting of light-activated *mouse* melanopsin (mOpn4L, targeted into 5-HT$_{2A}$ receptor domains[20,35]) to trigger G$_q$-signaling (Fig. 1g–i; Extended Data Figs. 1 and 2, Supplementary Movies 1–5) that mimics 5-HT$_{2A}$ receptor activation in V1 of anesthetized NEX-Cre and PV-Cre mice (Fig. 1e, f, Extended Data Figs. 3 and 4). To allow dense spatial sampling with good isolation and classification of simultaneously recorded single units as putative excitatory or inhibitory neurons (Fig. 1d), we

performed recordings of extracellular activity using multi-channel silicon probes.

## Cell-type-specific modulation of spontaneous activity by activation of the 5-HT$_{2A}$ receptor pathway

How is spontaneous activity affected when the 5-HT$_{2A}$ receptor pathway is activated in a cell-type-specific manner? To address this question, we isolated the modulatory effect of 5-HT$_{2A}$ signaling across distinct subpopulations of neurons (note that our recordings contain a

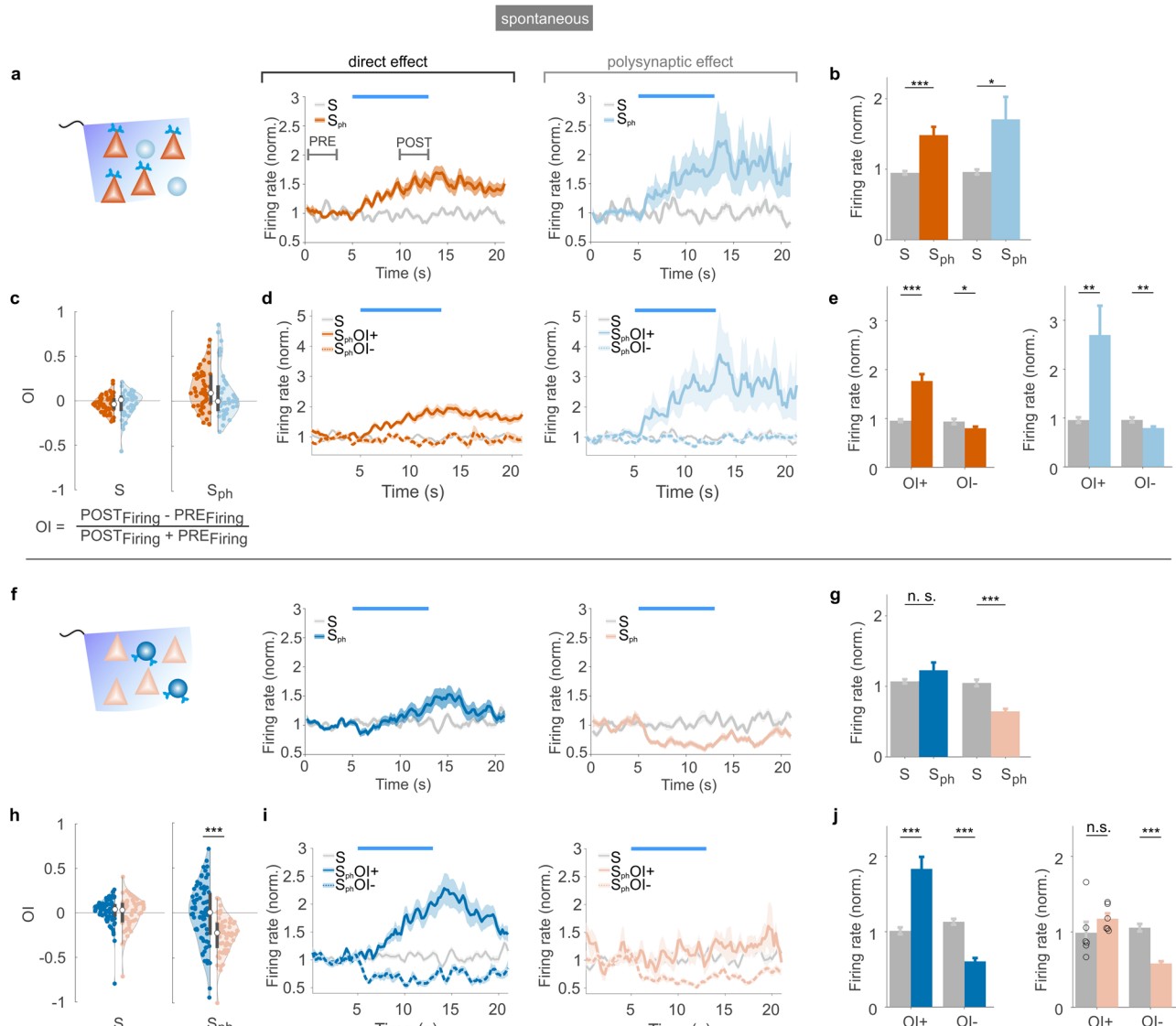

**Fig. 2 | Cell-type-specific activation of the 5-HT$_{2A}$ receptor pathway and modulations of spontaneous activity. a** Photostimulation of the 5-HT$_{2A}$ receptor pathway in pyramidal neurons. Left: Scheme of paradigm. Middle: spontaneous activity of excitatory neurons upon photostimulation (S$_{ph}$, dark orange), and under control condition (S, gray (throughout figure)). Blue bar shows the photostimulation time. Right: same conditions for the recorded pool of inhibitory neurons. Data represent mean ± SEM (shadings) of $n = 55$ excitatory units and $n = 44$ inhibitory units in 11 NEX-Cre mice. **b** Quantification (mean, error bars show + SEM) of the normalized firing rates in (**a**) in the time interval of 10–13 s (marked as 'POST' in **a**), (color scheme as in **a**). **c** Violin plots of opto-index (OI, see formula) values of all excitatory (dark orange circles) and inhibitory (light blue circles) neurons presented in (**a, b**), with kernel density estimation for the two populations; the left plot shows values for controls using OI-congruent PRE and POST times. **d** Activity of excitatory neurons (dark orange) with positive OI (solid line, S$_{ph}$OI+, $n = 39$) and

negative OI (stippled line, S$_{ph}$OI−, $n = 16$) and activity of inhibitory neurons (light blue; S$_{ph}$OI+, $n = 21$, S$_{ph}$OI−, $n = 22$). Data represent mean ± SEM (shadings). **e** Data in (**d**) quantified as in (**b**) (color scheme as in **d**). **f** Photostimulation of the 5-HT$_{2A}$ receptor pathway in PV interneurons. Middle: pool of inhibitory neurons during control condition (S, gray) and following photostimulation (S$_{ph}$, dark blue). Right: pool of excitatory neurons recorded under same conditions, control (gray) and with photostimulation (light orange). Data represents mean ± SEM (shadings) of $n = 73$ inhibitory units (37 with OI+, 36 with OI−) and $n = 56$ excitatory units (6 with OI+, 50 with OI−) recorded in 14 PV-Cre mice. **g** Quantification of normalized firing rates in (**f**) (averaged over same time interval as in **a**). (**h, i, j**) Same analysis and conventions as in (**c–e**) for data shown in (**f**). ***$p < 0.001$, **$p < 0.01$, and *$p < 0.05$, two-sided paired sample t-test in (**b** and **g**), one-sided in (**e** and **j**) or two-sample Kolmogorov–Smirnov test in (**h**). Exact $p$-values of all comparisons are reported in the Source Data file.

mixture of directly and indirectly activated neurons). In comparison to the control spontaneous firing (S), optogenetic activation of the 5-HT$_{2A}$ receptor pathway in pyramidal neurons (S$_{ph}$, Fig. 2a–e) led to an increase in spontaneous activity of both excitatory (Fig. 2a, dark orange trace, "direct effect", +48% cf. Fig. 2b) and inhibitory neurons (Fig. 2a, light blue trace, indirect, "polysynaptic effect", +70% cf. Fig. 2b). The slow rises and sustained increase in activity reflect the population dynamics as typically observed for GPCRs such as melanopsin or 5-HT receptors[35]. To further quantify these effects, we calculated an opto-index (OI), which scales values between −1 (infinite suppression) and +1 (infinite increase), while 0 indicates no effect on the baseline. The OI probability density functions show a similar distribution for the two neuronal populations, both peaking at a positive OI value (Fig. 2c–e) and with only a weak correlation between OI and firing rates (Extended Data Fig. 5). Quantification of activity changes shows a clear increase in activity of neurons with positive OI (Fig. 2d, e, NEX-Cre, excitatory: +76 ± 13%, inhibitory: +169 ± 59%), while a more modest decrease in OI−negative neurons was observed (NEX-Cre, excitatory: −19 ± 3%, inhibitory: −20 ± 3%).

Next, we activated the 5-HT$_{2A}$ receptor pathway specifically in PV interneurons. We found a slight but not significant increase in the firing of inhibitory neurons (Fig. 2f dark blue trace, +12% cf. Fig. 2g). However, it should be taken into account that the recorded sample of inhibitory neurons represent a heterogeneous group of interneurons, characterized by a distribution of the OI that is symmetrically centered around 0 (Fig. 2h, i left panel). One population of interneurons most likely represents PV neurons, which increase firing due to photoactivation of the 5-HT$_{2A}$ receptor ("direct effect", see Fig. 2i solid dark blue trace, +83 ± 15% cf. Fig. 2j left panel) while subsequently suppressing other inhibitory neurons ("polysynaptic effect", see Fig. 2i stippled dark blue trace, −39 ± 4% cf. Fig. 2j left panel; Extended Data Fig. 6 shows an example of monosynaptically connected interneurons). We also noticed a significant decrease in local field potential (LFP) power across different frequency bands caused by 5-HT$_{2A}$ receptor activation in PV neurons (Extended Data Fig. 7).

Activation of the 5-HT$_{2A}$ receptor pathway in PV interneurons led to an overall decrease in spontaneous activity of excitatory neurons (Fig. 2f, light orange traces, −35% cf. Fig. 2g). The population of excitatory neurons shows a distribution with a clear negative average OI (Fig. 2h, i right panel, light orange data; excitatory neurons with negative OI: −41 ± 2% cf. Fig. 2j right panel) with only minor increase in activity of neurons with positive OI (17 ± 6% cf. Fig. 2j right panel). The overall decrease in spontaneous activity of excitatory neurons is most likely due to the hyperpolarization caused by PV neurons via photoactivation of the 5-HT$_{2A}$ receptor pathway. Altogether, photostimulation of the 5-HT$_{2A}$ pathway separately in PV neurons produces heterogeneous bidirectional effects (suppression and facilitation) in inhibitory neurons and causes suppression in excitatory neurons. Whereas photostimulation of the 5-HT$_{2A}$ pathway in pyramidal neurons activates both excitatory and inhibitory populations.

## 5-HT$_{2A}$ signaling in PV neurons controls visual gain in excitatory neurons

Next, we investigated the effect of cell-type-specific 5-HT$_{2A}$ activation on visually evoked responses. Visual stimuli consisting of vertical or horizontal gratings with 100% contrast were repeatedly presented at an interval of 3 s (Fig. 3). We measured neuronal activity during visual stimulation alone (Fig. 3a, b, black traces, V) and with concurrent photostimulation of the 5-HT$_{2A}$ receptor pathway (Fig. 3a, b, blue traces, V$_{ph}$). To obtain the photostimulation-related response component over time, we subtracted the two traces. The resulting activity (Fig. 3a, b, gray traces, V$_{ph}$-V) resembled the trends in spontaneous activity found during 5-HT$_{2A}$ photostimulation across all neuron types except when the 5-HT$_{2A}$ receptor pathway was stimulated in PV interneurons, and activity was measured in excitatory neurons (Fig. 3b, light

orange encircled panel). There was no effect on the OI magnitude when pyramidal neurons were separately photoactivated (Fig. 3c). Only photostimulation of PV interneurons caused excitatory neurons to display transient dips in their V$_{ph}$-V trace (cf. Fig. 3b, gray trace in the light orange encircled panel), indicating a substantially larger suppression of evoked activity than spontaneous activity during visual responses (Fig. 3d). In all other cases the changes in amplitude of the visual responses could be solely explained by 5-HT$_{2A}$-induced effects on baseline (cf. Fig. 2). Thus, a reduction in the gain of visually evoked activity was exclusively revealed in the response magnitude (i.e., the difference between response amplitude and baseline, Fig. 3e left sketch) of excitatory neurons after activation of the 5-HT$_{2A}$ receptor pathway in PV neurons (Fig. 3e, light orange trace) without changing preferred orientation[31] (Extended Data Fig. 8).

A linear regression model applied to the obtained magnitude values of all excitatory neurons revealed that gain suppression is divisive (16% reduction in slope, i.e., $\beta_4/\beta_2$ in Fig. 3f; see Extended Data Fig. 9 for similar results using various contrasts). We conclude that the divisive control of visual input is largely based on an "indirect" polysynaptic network effect triggered by "direct" 5-HT$_{2A}$ activation in PV interneurons.

## Simultaneous activation of 5-HT$_{2A}$ in pyramidal and PV neurons allows for gain control of external input without changing levels of baseline activity

Next, we activated the 5-HT$_{2A}$ pathway in both pyramidal and PV neurons simultaneously (see "Methods" section). This "systemic" activation revealed two substantial deviations from a simple superposition of the above results. First, in contrast to the above 5-HT$_{2A}$ pathway activation in the single subpopulations, no changes in baseline amplitude was observed after activating the pathway jointly across both cell types (Fig. 4a, Extended Data Fig. 10a). Second, gain control of visual input was no longer exclusively present in excitatory neurons (cf. Fig. 3e, f) but additionally found in inhibitory neurons (Fig. 4b, d, Extended Data Fig. 10b). Thus, we obtained a striking difference to a linear effect suggesting a significant change of the cortical processing regime when the 5-HT$_{2A}$ pathway is concurrently activated in pyramidal and PV neurons. Moreover, the maximal suppression in response magnitude of excitatory neurons was significantly stronger in the systemic condition than for 5-HT$_{2A}$ pathway activation in only PV neurons (54% vs 20%, $p < 0.01$, two-sided two-sample t-test).

Whether the observed divisive suppression (Fig. 4c, d) displays also properties of normalization remains to be tested with different stimulus contrasts. A previous study, in which the release of 5-HT was optogenetically controlled, showed magnitude normalization of V1 responses to varying contrasts, most likely involving 5-HT$_{2A}$ signaling as suggested by additional pharmacological manipulations[29].

## Spiking network model suggests changes in cortical processing regime through 5-HT$_{2A}$-induced polysynaptic mechanism

We employed a spiking cortical network model developed by Sadeh et al.[36] with slight modifications, consisting of excitatory and inhibitory units driven by realistic neuronal parameters (Table 1). 5-HT$_{2A}$ activation was introduced to this model by adding background excitatory input to varying fractions of units. This free parameter also accounts for the fact that PV neurons are a subpopulation of inhibitory neurons and for variations in the expression strength of our 5-HT$_{2A}$ viral construct.

Similar to the measured neuronal data, when activating 5-HT$_{2A}$ in only pyramidal cells, the model showed increased spontaneous activity of both inhibitory and excitatory units (Fig. 5a, left data panel, Extended Data Fig. 11a), demonstrating an overall excitatory effect on the network (Extended Data Fig. 12a). Conversely, separate 5-HT$_{2A}$ activation of inhibitory units caused an increase of their baseline activity (Fig. 5b, left data panel, dark blue), while baseline activity of

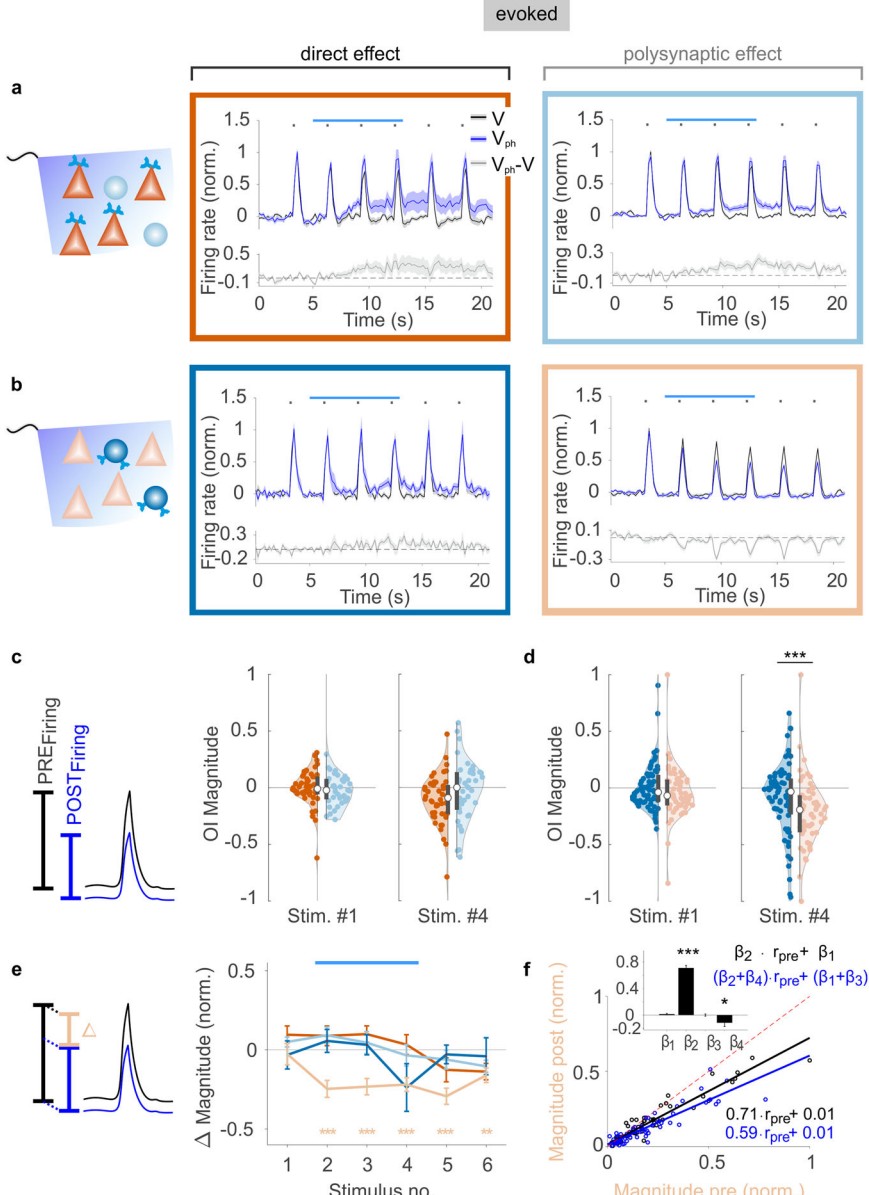

**Fig. 3 | Cell-type-specific activation of the 5-HT$_{2A}$ receptor pathway and modulations of evoked activity. a, b** Visual responses (dots indicate visual stimulus timing) in control conditions (V, black) and with additional 5-HT$_{2A}$ receptor activation (V$_{ph}$, blue; horizontal bars). Gray traces represent (V$_{ph}$-V).
**a** Photostimulation of the 5-HT$_{2A}$ pathway in pyramidal neurons. Normalized evoked activity of excitatory and inhibitory neurons (dark orange and blue box, respectively), mean ± SEM (shadings) of $n = 55$ excitatory units and $n = 44$ inhibitory units 11 in NEX-Cre mice. **b** Photostimulation of the 5-HT$_{2A}$ pathway in PV interneurons. Normalized evoked activity of inhibitory and excitatory neurons (dark blue and light orange box, respectively), mean ± SEM (shadings) of $n = 73$ inhibitory units and $n = 56$ excitatory units recorded in 14 PV-Cre mice. **c** Violin plot of opto-index (OI) based on response magnitude (difference between peak amplitude and baseline, see scheme at left) for excitatory (dark orange circles) and inhibitory (light blue circles) neurons in (**a**). Compared are values for the control visual stimulus #1 (pre-photostimulation) and stimulus #4 (during photostimulation; x-axis in (**e**)

shows stimulus numbering). **d** Violin plot of the same analysis as in (**c**) for data shown in (**b**). ***$p = 0.0007$, two-sample Kolmogorov–Smirnov test.
**e** Quantification of response magnitude of the conditions shown in (**a, b**). Color scheme denotes cell type and is matched to box colors above. ***$p < 0.00017$, **$p < 0.0017$, two-sided paired sample t-test with Bonferroni correction; exact p-values of comparisons are reported in the Source Data file. **f** Comparison between response magnitude evoked by stimulus #1 and the average of magnitude values obtained for stimuli #2–4 for each excitatory unit during optogenetic activation of 5-HT$_{2A}$ in PV interneurons (see **b**, right). Control (V, black circles), photostimulation conditions (V$_{ph}$, blue circles), data normalized to the unit with the highest firing rate ($n = 58$). Lines represent linear regression for V and V$_{ph}$, identity line (dashed red). Inset represents regression coefficients (mean ± SEM, $n = 116$). The negative value of the coefficient $\beta_4$ indicates divisive reduction of the magnitude in the V$_{ph}$ condition. ***$p = 6.09 \times 10^{-37}$ ($\beta_2$), *$p = 0.044$ ($\beta_4$), two-sided one-sample t-test. Source data are provided as a Source Data file.

excitatory units was suppressed (Fig. 5b, left data panel, light orange; Extended Data Fig. 12b). Moreover, the OI of directly stimulated inhibitory units strictly increased, while the OI of indirectly affected inhibitory units decreased (Extended Data Fig. 11b). In contrast, the solely, indirectly stimulated excitatory subpopulation maintains a fairly compact OI distribution (Extended Data Fig. 11b). This matches the

experimental data (Fig. 2h) where the shape of the OI distribution of the excitatory population remains roughly similar to the control condition while the distribution of the inhibitory population becomes much broader following photostimulation. Thus, the model captured both the experimentally observed depolarizing effect within subpopulations of interneurons via the direct 5-HT$_{2A}$ pathway and the

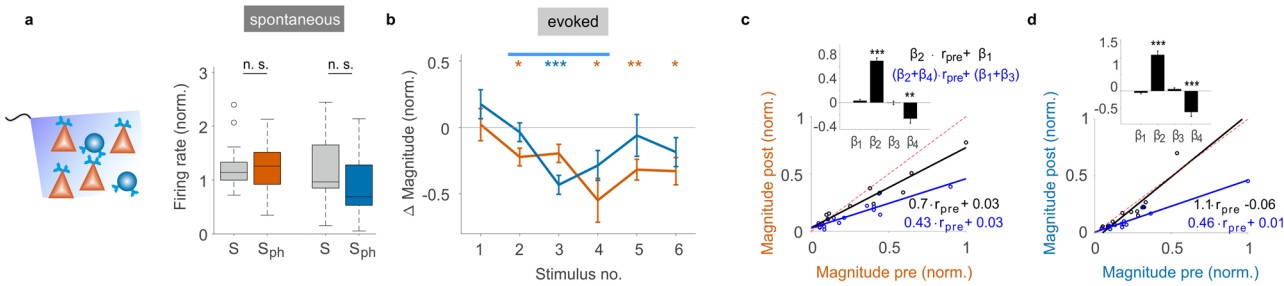

**Fig. 4 | Systemic activation of the 5-HT$_{2A}$ pathway in pyramidal and parvalbumin neurons stabilizes spontaneous activity levels and controls visual input gain. a** Left: Scheme of paradigm. Right: Quantification of the normalized spontaneous firing rates, dark blue representing inhibitory neurons ($n = 13$) and dark orange excitatory neurons ($n = 14$), mean recording depth 318 μm ±78 (SD) in 5 PV-Cre mice (for average time traces see Extended Data Fig. 10). Box plots indicate median (middle line), 25th, 75th percentile (box), ±2.7 sigma (whiskers), outliers are plotted as 'o'; ns not significant, two-sided paired sample t-test with Bonferroni correction. **b** Quantification of response magnitude for each visual stimulus (calculated as in Fig. 3e). Data represent mean, error bars show ± SEM. Color scheme denotes cell type. ***$p < 0.00017$, **$p < 0.0017$, *$p < 0.0083$, two-sided one-sample t-test with Bonferroni correction for multiple comparisons; exact $p$-values are

reported in the Source Data file. **c** Comparison of response magnitude before photostimulation (V, visual stimulus #1, black circles) and during photostimulation (V$_{ph}$, average across visual stimuli #2–4, blue circles) for each excitatory unit in (**a**, **b**), regression equations are depicted with corresponding colors, lines represent linear regression for V and V$_{ph}$ (dashed red line represents identity line). Data was normalized to the unit with the highest firing rate. Inset represents regression coefficients (mean ± SEM, $n = 28$). The negative value of the coefficient β$_4$ indicates significant divisive reduction of the magnitude in the V$_{ph}$ condition. ***$p = 3.78 \times 10^{-13}$ (β$_2$), **$p = 0.0018$ (β$_4$), two-sided one-sample t-test. **d** same as (**c**) for all inhibitory units in (**a**, **b**) (mean ± SEM, $n = 26$, ***$p = 2.19 \times 10^{-9}$ (β$_2$), ***$p = 7.33 \times 10^{-5}$ (β$_4$), two-sided one-sample t-test. Source data are provided as a Source Data file.

### Table 1 | Neural dynamics parameters used in neural network simulations

| Parameter | Symbol | Value |
|---|---|---|
| Membrane capacitance | $C$ | 120 pF |
| Reversal potential of excitatory conductances | $E_e$ | 0 mV |
| Reversal potential of inhibitory conductances | $E_i$ | −75 mV |
| Resting potential | $E_L$ | −70 mV |
| Spiking threshold voltage | $E_T$ | −50 mV |
| Leak conductance | $g_L$ | 7.14 nS |
| Refractory period | $t_{ref}$ | 2 ms |
| Synaptic time constant (excitatory input) | $\tau_e$ | 1 ms |
| Synaptic time constant (inhibitory input) | $\tau_i$ | 1 ms |
| Reset voltage | $V_r$ | −60 mV |

"indirect" (i.e., multi-synaptic) effect, that is, their inhibitory influence on excitatory units. The network model also reproduced the experimentally observed changes in visually evoked responses. Separate 5-HT$_{2A}$ activation in the subpopulation of excitatory units led to minor rises in evoked magnitude across the entire network (Fig. 5a, right panel). Conversely, separate 5-HT$_{2A}$ activation in inhibitory units led to a slight decrease in response magnitude across the population of inhibitory units (Fig. 5b, right panel, dark blue). However, magnitude gain was most prominently reduced in excitatory units (Fig. 5b, right panel, light orange) and also divisive (Extended Data Fig. 13), similar to our physiological data (Fig. 3).

Finally, we tested the model's prediction for systemic activation of 5-HT$_{2A}$ receptors, mimicking their joint activation across excitatory and inhibitory neurons. When modeling the above effects in a mean-driven regime where all parameters are constant, yielded approximately an additive effect, the sum of activating 5-HT$_{2A}$ in each subpopulation (cf. Fig. 5a, b) separately. Even though excitatory units displayed negligible changes in baseline firing as in the experiments, significant modulation of evoked response gain was absent in these units (Fig. 5c, triangles). Contrary to the systemic activation in the experiments, the baseline activity in the inhibitory units increased without any effect on response gain (Fig. 5c, circles). Hence, we observed different results for the mean-driven regime (Fig. 4) and therefore, the simple linear model substantially fails in explaining the experimentally revealed values.

When all other parameters remain constant, increasing 5-HT$_{2A}$-mediated excitatory and inhibitory conductance led to increases in baseline activity. Whereas at higher levels of conductance, baseline activity declined. Indeed, previous studies applying 5-HT$_{2A}$ agonist in monkey visual cortex showed bidirectional modulatory effects on visual responses dependent on initial firing rates[27]. However, recent pharmacological studies[29,31] that employed systemic modulation of endogenous 5-HT$_{2A}$ receptors in mice (targeting 5-HT$_{2A}$ simultaneously across all cell types) suggested that spontaneous activity levels are unaffected. In these two studies, systemic activation of the 5-HT$_{2A}$ receptor had strong suppressive effects on evoked visual responses (−21 and −67%, respectively).

How is it possible then, that following systemic and specific 5-HT$_{2A}$ activation, the baseline firing rate remains constant, while at the same time, response amplitudes are modulated? To reconcile our present findings, we consider that our network model operates in a fluctuation-driven regime[37]. In this regime, the mean membrane potential of a given unit does not change while both excitatory and inhibitory input rates increase, i.e., by balancing each other[38,39]. However, the fluctuations of the membrane potential change in a non-monotonic, inverted U-shaped, fashion[37]. For low conductance, fluctuations increase with the conductance, which leads to an increase in the firing rate of a target neuron as fluctuations exceed the spiking threshold more often. For high conductance, the fluctuations decrease, and thus, the firing rate of the target unit decreases. The model correctly predicts baseline stability and visual gain suppression following systemic 5-HT$_{2A}$ activation (Fig. 5d). Moreover, it reproduced that visual gain suppression is stronger in systemic activation than for isolated PV-specific 5-HT$_{2A}$ activation (cf. Fig. 5b, light orange trace and Fig. 5d, blue trace with triangles in the right panel). Our systemic activation data and the previously reported values of visual gain suppression[29,31] are in agreement with the fluctuation-driven model when 35–85% of all excitatory and inhibitory units are activated by 5-HT$_{2A}$ receptors (Fig. 5d, blue area). Thus, modeling of systemic 5-HT$_{2A}$ receptor activation in a balanced regime (Extended Data Fig. 14) shows that the network maintains a stable baseline, buffering against spontaneous activity changes, while modulating external input gain.

Specifically, the inverted U-shape profile of conductance-dependent membrane fluctuations enables units in the model to exhibit the same baseline firing rate at two different rates of excitatory and inhibitory synaptic inputs (Fig. 5e, two left panels). The low-rate

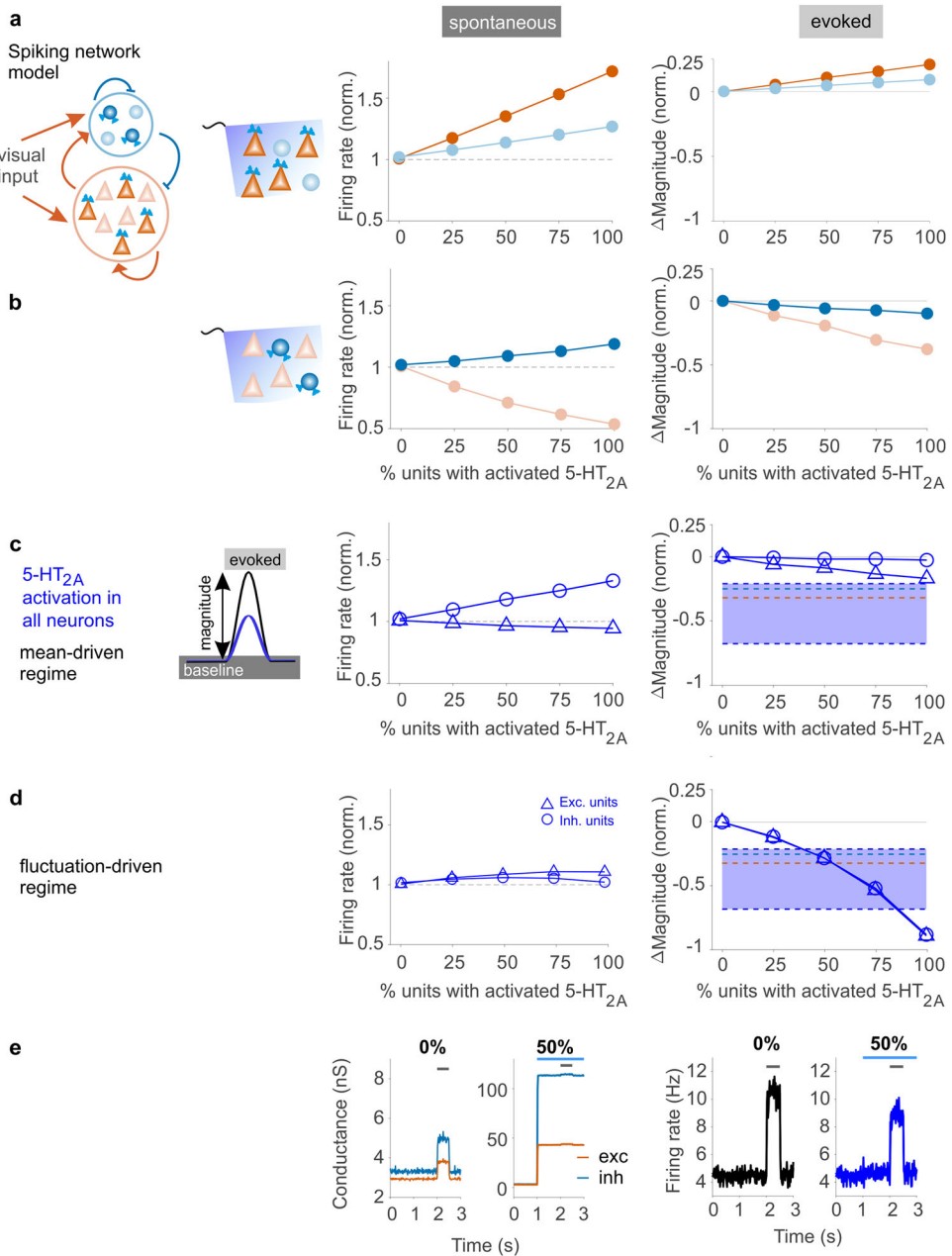

**Fig. 5 | Cortical network model predicts 5-HT$_{2A}$ receptor-induced modulations in activity. a** Left: Schematics of the spiking network model showing interactions between different pools of neurons (excitatory, inhibitory), activation of the 5-HT$_{2A}$ receptor, and visual input. 5-HT$_{2A}$ receptor activation in only excitatory units revealed changes in spontaneous firing (middle) and stimulus-evoked magnitude (right) for excitatory units (dark orange) and inhibitory units (light blue) dependent on the percentage of excitatory units with activated 5-HT$_{2A}$ receptors. **b** 5-HT$_{2A}$ receptor activation in only inhibitory units. Same analysis as in (**a**). Color saturation in (**a**, **b**) indicates that the 5-HT$_{2A}$ receptor was activated in the analyzed type of unit ("direct effect", dark colors) or instead affects the analyzed unit via (poly-) synaptic contacts ("indirect network effects", light colors,); color scheme as in Figs. 2 and 3. **c** Mean-driven regime: Predicted effects of "systemic" activation (simultaneous activation of 5-HT$_{2A}$ in both excitatory and inhibitory units) on baseline firing (middle) and visual response magnitude (right) assuming linear superposition of the model values shown in (**a**, **b**). Blue shaded area marks the range of suppression of visual gain observed for systemic activation in the current study (−32% and −25% (average across stim #2–4) for excitatory (orange stippled line) and inhibitory neurons (light blue stippled line), respectively) and of presumably 5-HT$_{2A}$ receptor-induced suppression reported in previous studies (−21%[29] and −67%[31], dark blue stippled lines). **d** Same as (**c**) for the fluctuation-driven regime. **e** Left: Average excitatory (orange) and inhibitory (blue) conductance predicted by the model with no 5-HT$_{2A}$ activation (0%) and during systemic 5-HT$_{2A}$ receptor activation in 50% of all cells (please note the different Y scales). Visual stimulation time is indicated by the black bar, blue marks receptor activation. Right: Firing rate with no 5-HT$_{2A}$ stimulation (0%) and during systemic 5-HT$_{2A}$ activation in 50% of all cells. Note that the increase in conductance following 5-HT$_{2A}$ activation leads to suppression of the response to visual input without changes in spontaneous firing rate. Source data are provided as a Source Data file.

setting mimics input rates without 5-HT$_{2A}$ activation (Fig. 5e, 0%), while the high-rate setting mimics the input rates when the 5-HT$_{2A}$ receptor is activated systemically (Fig. 5e, blue bar indicates 5-HT$_{2A}$ receptor activation, 50% of all units in the example shown). As a result, spontaneous spiking remains constant in both cases (Fig. 5e, compare pre-stimulus times, 0–2 s in the two right panels) while the sensory gain is reduced (Fig. 5e, compare amplitudes of black (0%) and blue (50%) traces, respectively).

## Discussion

Our experimental and modeling results suggest that the 5-HT$_{2A}$ receptor contributes to divisive control of response gain. Its activation in PV interneurons directly mediates this effect, while its simultaneous activation in pyramidal cells may further amplify suppression of response gain through increased overall conductance. The 5-HT$_{2A}$ receptor can separately modulate spontaneous activity upon cell-specific activation but has little net baseline effect during strong systemic drive. Hence, at the network level, the 5-HT$_{2A}$ receptor supports specific and independent modulation of one activity stream, i.e. visually evoked input, while leaving the other one, i.e., spontaneous ongoing activity, largely intact. Whether these findings also hold true for cortices of other modalities needs further exploration.

Further we showed that selective optogenetic activation of the 5-HT$_{2A}$ receptor pathway in distinct cortical populations, either pyramidal or PV neurons, uncovers a mixture of direct (i.e. receptor activated) and indirect (i.e. postsynaptic) effects of increased and decreased spiking dependent on neuron type. Combining these multiple effects in a computational model revealed that the systemic activation of cortical 5-HT$_{2A}$ receptors results in increased overall conductance across the network. As a consequence, the reduced driving force of sensory input produced suppression of evoked visual spiking responses. Thus, our model explains the 5-HT-induced reduction in pyramidal cell output upon visual stimulation[27–31] by a reciprocal balanced action of the 5-HT$_{2A}$ receptor in antagonistic (excitatory vs. inhibitory) neuron types.

Importantly, we found that tonic balanced excitatory and inhibitory drive by 5-HT$_{2A}$ activation increased conductance across the network with only mild net effects on the average ongoing spiking activity. This indicates functional segregation in which 5-HT$_{2A}$ activation produces a coherent and selective impact on sensory input[40]. In addition, the tonic 5-HT$_{2A}$-mediated increase in baseline conductance was much higher than the increase in conductance elicited by visual input. This suggests that sensory gain modulation comes at the cost of high metabolic turnover when 5-HT levels are elevated. Interestingly, 5-HT levels change during day-night cycle with lower values during rest[41,42]. We, therefore, speculate that reduced 5-HT levels partly contribute to lower energy consumption and cortical resting-state changes during sleep, when the processing of sensory stimuli is naturally reduced compared to awake state.

Our computational model potentially solves two apparently conflicting observations. First, during systemic activation, the gain of the visually evoked response is suppressed even though the baseline firing rate is unchanged. Second, during cell-type-specific modulation both spontaneous and visually evoked response change. Using a single set of modeling parameters, by modulating 5-HT$_{2A}$–mediated changes in conductance, we were able to account for all experimental results. However, our model crucially depends on the network operating in a fluctuation-drive regime[37]. Although visual cortical neurons have been shown to operate in this regime during sensory processing[43], further details of the modeling constraints remain to be explored. The reduction in gain obtained here *in-silico* were similar to our experimental findings and values in the literature[29,31]. However, the involvement of other 5-HT receptor- and cell types, most likely contributing to a further fine-tuning of network responses should be considered[15,27,44–49]. For example, the expression pattern of our construct does not concur with the normal complex distribution of 5-HT$_{2A}$ receptors across cortical layers[47], which naturally serves further signal tuning within a spectrum of functions. Thus, the dependence of the mechanisms on layer-specific circuitries needs further study. Using subcutaneous injections of 2,5-dimethoxy-4-iodoamphetamine (DOI) to stimulate endogenous 5-HT$_{2A}$ receptors in mice, Michaiel and colleagues[31] showed that suppressive effects were more pronounced in layer 2/3 and also cell type-specific. Moreover, the expression level of optogenetic tools can vary, as accounted for in our model by varying the percentage of activated units (Extended Data Fig. 14). Note also that our recordings include neurons with different preferred orientations. Thus, our visual stimulation with a single orientation yields an average effect of gain suppression over the population rather than scrutinizing effects on individual tuning curves. Hence, possible mixtures of divisive and subtractive or purely subtractive suppression at the single neuron level[8] might be undetected. Interestingly, similar to our observations, Michaiel and colleagues[31] found no changes in preferred orientation following DOI administration. Finally, the impact of 5-HT signaling may differ under wakeful conditions compared to the anesthetized state where inhibition and 5-HT levels are affected[50,51]. In fact, we showed recently that 5-HT-induced suppressive effects are less pronounced under awake conditions as compared to anesthetized preparations[29]. However, while the value of effect sizes can deviate between the light-activated and the native 5-HT$_{2A}$ system, it appears unlikely that our manipulation imposes fundamental different operations across the native functional architecture. Our findings demonstrate that joint and balanced activation of the 5-HT$_{2A}$ pathway (i.e. simultaneous G$_q$-signaling in pyramidal and PV neurons) leads to a qualitative change in the cortical state which allows for gain control of external input without changes in baseline.

We conclude that 5-HT$_{2A}$-induced gain modulation of external input is achieved through balanced excitatory and inhibitory ongoing synaptic input[52,53] with only minor effects on the amplitude of baseline activity. This allows independent control of the weight of external sensory input and ongoing internal communication processes[29]. Modulation of 5-HT$_{2A}$ receptor contribution[54] may permit flexible segregation[55] and integration[56] of ongoing activity (including top-down feedback[57,58]) to achieve context-dependent scaling of input. This also supports the notion that these functions are sensitive and prone to malfunction when imbalances occur in the distribution or activation of 5-HT$_{2A}$ receptors across neuron types[59–62]. Altogether our results shed light on network mechanisms of gain control by modulatory systems, influencing sensory impact on cortical dynamics, and providing distinct control of various streams of information via GPCRs.

## Methods

### Animals

Adult male and female heterozygous NEX-Cre[63] and homozygous PV-Cre[64] mice were used in this study to induce expression of the injected construct in either pyramidal neurons or parvalbumin (PV) interneurons. All experimental procedures were carried out in accordance with the European Union Community Council guidelines and approved by the German Animal Care and Use Committee under the Deutsches Tierschutzgesetz (Az.: 84-02.04.2014.A439, 84-02.04.2016.A138, 84-02.04.2019.A483, 81-02.04.2019.A228, 81-02.04.2021.A412) and the NIH guidelines. Mice were group-housed in standard vivarium conditions (temperature maintained at 20–22 °C, 30–70% humidity). After stereotactic injections (detailed below), mice were housed individually, in a 12 h light/dark cycle with food and water ad libitum.

### Generation of plasmid constructs

Adeno-associated viral (AAV) vector construction of pAAV-CMV-floxed-mOpn4L-eGFP-5-HT$_{2A}$CT was based on pAAV-eGFP-CW3SL (GenBank accession number: KJ411916.2) as a backbone plasmid, in which loxP and lox2272 were introduced to create a conditional construct. Sequences for long isoform *mouse* melanopsin (mOpn4L[35], GenBank accession number: NM_013887.2), eGFP, and the C-terminus of human 5-hydroxytryptamine receptor 2A (5-HT$_{2A}$, amino acids K385-V471, GenBank accession number: NM_000621.4) were PCR amplified with 16 base pair overhangs and inserted into the backbone plasmid using In-Fusion cloning kit (Takara).

## Viral vector production

AAV stocks of serotype 2/9 were produced using the AAV helper-free system (Agilent Technologies, Santa Clara, CA). Three plasmids encoding (1) the DNA sequence of interest flanked by two inverted terminal repeats (ITRs), (2) the helper plasmid encoding AAV rep and cap genes, and (3) adenoviral helper plasmid were co-transfected in HEK293T cells using polyethyleneimine as a transfection reagent. After 72 h incubation, HEK293T cells and culture medium were harvested and centrifuged at low speed (300 × $g$, 10 min, 4 °C). Cells were resuspended in lysis buffer (150 mM NaCl, Tris–Cl pH 8.5) while the supernatant was incubated with 40% PEG-8000 at 4 °C on a shaker. Cells were lysed by 5 freeze-thaw cycles and incubated with DNase and MgCl$_2$ (30 min, 37 °C). Following centrifugation (3700 × $g$, 20 min, 4 °C), the supernatant from the cell lysate was used to resuspend PEG-8000 precipitates. For AAV purification, the resuspension was incubated with 40% PEG-8000 for 2 h, centrifuged (3700 × $g$, 20 min, 4 °C), the supernatant discarded, and the pellet resuspended in 10 ml 50 mM HEPES buffer. Room-temperature chloroform (1:1 volume) was added, the mixture thoroughly vortexed for 2 min, and centrifuged (370 × $g$, 5 min, RT). Finally, the aqueous phase was collected, sterile filtered (0.22 μm syringe filter membrane), and concentrated by incubation with 40% PEG-8000 for 2 h and subsequent centrifugation (3700 × $g$, 20 min, 4 °C). The resulting AAV stock was resuspended in 100 μl of 1× PBS with 0.001% Pluronic F68, aliquoted, and stored at −80 °C.

## Viral construct and stereotactic injections

The viral solution of AAV containing the 5-HT$_{2A}$ receptor construct, pAAV-CMV-floxed-mOpn4L-eGFP-5-HT$_{2A}$CT, was injected in the *mouse* primary visual cortex. In order to express the construct simultaneously in both PV and pyramidal neurons, a mixed viral solution containing pAAV-CMV-floxed-mOpn4L-eGFP-5-HT$_{2A}$CT and pAAV-CamKII(0.4)-mOpn4L-eGFP-5-HT$_{2A}$CT was used and injections were performed in PV-Cre mice (Extended Data Fig. 15). Animals were anaesthetized with isoflurane (3% induction and 1–1.5% maintenance) via a nose mask. Animals were placed in the stereotactic frame on a heating pad (38 °C) and injected subcutaneously with 0.2–0.25 ml NaCl (0.9%) solution containing buprenorphine (10 μg/ml; 0.1 μg/g bodyweight) and atropine (5 μg/ml; 0.05 μg/g bodyweight). Lidocaine cream (2%) was applied on the scalp for local anesthesia. After the scalp was cut with a circular incision, the edges of the skin were glued to the sides of the skull with histoacryl glue. The dorsal part of the skull was thinned using a microdrill until the vascular structure was clearly visible. Two small craniotomies (~100 μm diameter) were made in each hemisphere above the visual cortex based on stereotactic coordinates (from Bregma: 4.2 mm posterior and 2.5 mm lateral; 3.5 mm posterior and 2 mm lateral). These coordinates were further adjusted beforehand for each animal, by multiplying them with a factor calculated as the ratio between the animal's Bregma-Lambda distance and the default Bregma-Lambda distance (4.2 mm). Using a micromanipulator and a glass pipette coupled to a syringe via a silicone tube, the virus was injected into the craniotomies in three steps, at depths of 600, 400, and 200 μm, with 10 min breaks between the steps. After the injections, the skull was covered with transparent dental cement (Super-Bond C&B Set) for protection and a transparent nail polish layer to reduce light scattering. A head holder was attached using dental cement to provide stable mounting of the animals during experiments. A green silicon elastomer (Kwik-Cast™, WPI) was placed above the nail polish layer to avoid photoactivation of the optogenetic probe by ambient light.

## Visual stimuli and photostimulation

A monitor (60 Hz, mean luminance 55 cd/m$^2$, LG 24BK55WV-B) was placed 30 cm away from the stimulated eye, covering ~56 × 70 deg of the visual field. A semipermeable zero-power contact lens was used to prevent the eye from drying out. The non-stimulated eye was covered with eye cream (Visco-Vision gel, OmniVision®) and aluminum foil to prevent any input by ambient light. Each experiment consisted of 10–25 trials and included four types of conditions, presented in a block-randomized order: (1) S – spontaneous condition, during which a uniform isoluminant gray screen was presented, (2) V – visually evoked condition (control), during which moving gratings were presented, (3) S$_{ph}$ – spontaneous condition, during photostimulation, inducing 5-HT$_{2A}$ receptor signaling, and (4) V$_{ph}$ – visually evoked condition during photostimulation of 5-HT$_{2A}$ receptors. Visual stimuli consisted of vertical or horizontal square-wave gratings (0.05 cycles/deg), moving at 2 cycles/s. Stimuli were repeatedly presented (6 times at a 3 s inter-stimulus interval) for 200 ms. In photostimulation conditions, continuous light (465 nm, applied with LED driver and module system, Plexon) lasted 8 s, which in the V$_{ph}$ condition started 1.8 s after the 1st visual stimulus onset. The time interval between two consecutive trials was 40 s. Light was delivered via an optic fiber connected to the silicon probe ("optrode", Cambridge NeuroTech). The fiber ended 750 μm above the electrode tip, light power at the fiber tip was 1–1.5 mW. After the photostimulated trials, a continuous yellow light pulse (590 nm, LED driver and module system, Plexon) lasting for 30 s was delivered through the same optic fiber to switch off the opsin.

## In vivo extracellular recording

Extracellular single-unit activity (SUA) was recorded using acutely inserted silicon probes with a varying number of recording channels: 4-channel (Q1x1-tet-5mm-121-Q4, NeuroNexus Technologies), 16-, 32- or 64-channel silicon probes (ASSY-1 E-1, ASSY-1 P-1, ASSY-37 E-1, ASSY-37 P-1, ASSY-77 H6, Cambridge NeuroTech) coupled to the optic fiber. Signals were amplified with an amplifier board (RHD2132 or RHD2164, Intan Technologies) and recorded at 20 kHz sampling rate using a multi-channel electrophysiology acquisition board (Open Ephys[65]). For spike isolation, the signal was filtered between 600 to 6 kHz. Isolated spikes were detected and clustered using the Klusta suite[66] (https://github.com/klusta-team/klustakwik) or the Kilosort2 program[67,68] with default configuration parameters. Extracellular recordings, visual- and photo-stimulations were synchronized using a Master-8 Pulse Stimulator (A.M.P.I.) and Raspberry Pi devices, and custom-written scripts in Python.

## 2-Photon calcium imaging

NEX-Cre mice were injected as described above with viruses containing floxed constructs of mOpn4L-mCherry-5-HT$_{2A}$CT and GCaMP. After 10–14 days the mice were anaesthetized and the brains were extracted. Coronal slices (250 μm thick) including V1 were prepared in a solution containing 87 mM NaCl, 2.5 mM KCl, 0.5 mM CaCl$_2$, 7 mM MgCl$_2$, 1.25 mM NaH$_2$PO$_4$, 25 mM NaHCO$_3$, 10 mM D-glucose and 75 mM sucrose. The solution was bubbled with 95% O$_2$ and 5% CO$_2$ using the VT1000S Vibratome (Leica). All slices were consecutively incubated in 1 μM TTX (Tocris), 1 μM CNQX (Tocris) and 25 μM 9-cis retinal (Sigma) for 1 h at 37 °C in a solution consisting of 125 mM NaCl, 2.5 mM KCl, 2 mM CaCl$_2$, 1 mM MgSO$_4$, 1.25 mM NaH$_2$PO$_4$, 26 mM NaHCO$_3$, and 20 mM D-glucose, bubbled with 95% O$_2$ and 5% CO$_2$. Finally, the PLC antagonist U73122 (10 μM) was added to the bath solution to block the G$_q$ pathway. Calcium imaging was performed in 3 slices with a 2-photon imaging system (Bruker) using a 965 nm tunable laser and with 16× objective and 1.6× optical zoom. Prior to the recording of each condition (3 min), a 1100 nm tunable laser was used to set the focus level and to screen for mOpn4L-mCherry-5-HT$_{2A}$ expression. To exclude initial melanopsin activation, each slice was incubated for 10 min after setting the focus level. Calcium signals were detected using photomultipliers and fluorescent levels were quantified using the ImageJ Fiji software and the Time Series Analyzer V3 Plugin. For each cell, the fluorescent trace was normalized to its respective starting value (F$_O$), and mean ± SEM was calculated over all conditions and across all cells. Amplitudes of the calcium signals were quantified

between 7.2 s and 7.9 s after recording onset (around the time when maximal amplitudes were observed).

Cells were categorized by visual inspection as "inactive" if no response occurred during the recording (180 s), as "spontaneously active" if the activity was not directly locked to photoactivation, determined by activity occurring later than 20 s or as "light-activated" if the activity occurred within the first 20 s of recordings. For each condition, the fraction of cells that was activated by light (<20 s) and cells that were not photoactivated was quantified (Extended Data Fig. 1).

Hek tsA 201 cells were seeded in 35 mm culture dishes 48 h and transfected 24 h prior to the experiment with either Opn4L-5-HT$_{2A}$CT or native 5-HT$_{2A}$ receptor co-transfected with GCamP6m respectively and kept in the dark at all times. The Imaging was performed in regular DMEM high glucose medium (Gibco). For PLC deactivation experiments the dishes were preincubated with 10 μM of the PLC antagonist U73122 for 30 min. Calcium imaging was performed with the same 2-photon setup (Bruker) using the 965 nm tunable laser, a 40× objective, and no further optical zoom. Focusing cell surface levels beforehand was done using red light to exclude prior melanopsin activation. To initialize 5-HT$_{2A}$ receptor signaling 50 μM serotonin was applied after 30 s. Data Analysis was performed using the *Fiji Time Series Analyzer* Plugin. Absolute fluorescent values were measured for each cell, baseline values subtracted and normalized (Extended Data Fig. 2d).

## GsX assay

Hek tsa 201 cells were seeded into Poly-L lysin coated 96-well plates 48 h and co-transfected with the GloSensor (ProMega), Opn4L-5HT$_{2A}$CT or 5-HT$_{2A}$ receptor, and the respective G protein chimera for Gs, Gi, Go or Gq 24 h prior to the GsX assay[69,70] (transfection ratio 100:100:1). 1 h before the experiment cell culture medium is replaced by L-15 medium containing 2 mM beetle luciferin (and 1 μM 9-cis retinal in case of the Opn4L-5HT$_{2A}$CT assay). Luminescence recordings were performed using a PerkinElmer 2030 multilabel reader VICTOR X3. After 160 s baseline recordings, a stimulation period of 5 s blue LED light stimulation (Opn4L-5HT$_{2A}$CT) or application of 20 μM Ro 60-175 Fumarat (5-HT$_{2A}$ receptor) followed. Subsequently, luminescence was recorded every ~40 s for 1300 s. For Data Analysis baseline values were subtracted to correct for differences in expression levels. Mean fluorescence values were normalized to the maximal luminescent increase for each condition, Opn4L-5HT$_{2A}$CT and 5-HT$_{2A}$ receptor, respectively (Extended Data Fig. 2a–c).

## Immunohistochemistry

To check the specificity of the virus expression, the animals were perfused at the end of each experiment and immunohistochemical stainings were performed. The animals were anaesthetized with a mixture of Ketamine (0.1 mg/g bodyweight) and Xylazine (0.01 mg/g bodyweight) injected intraperitoneally (i.p.). After the exclusion of any signs of pain, transcardial perfusion of PBS (pH = 7.4) followed by 4% PFA was performed. The brains were removed, fixed overnight at 4 °C in 4% PFA, and cryoprotected in PBS solution with 30% sucrose. The brains were frozen in Tissue-Tek (O.C.T.™ Compound) and sectioned using a cryotome (Leica CM3050 S). Coronal slices (30 μm thick) were immersed in TBS on a 24-well plate (CytoOne®) for further immunohistochemical staining. All the incubation and rinsing steps took place on an orbital shaker and at room-temperature unless stated otherwise. Free-floating sections were rinsed 3 times (10 min) with TBS. Sections were incubated in a blocking solution containing 0.3% Triton-X-100 and 10% normal donkey serum (NDS) in TBS for 1 h, followed by primary antibody (1:300 rabbit anti-GluR2/3, 1:500 rabbit anti-GABA or 1:1000 *mouse* anti-PV) incubation in TBS containing 0.3% Triton-X-100 and 5% NDS for 1 day at 4 °C. After 3 steps of rinsing with TBS (10 min), the sections were incubated in the secondary antibody solution (1:500

donkey anti-rabbit DyLight650 and 1:500 donkey anti-*mouse* DyLight550 in 0.3% Triton-X-100 and 5% NDS in TBS) for 1.5 h while protected from light. The sections were then rinsed 3 times with TBS (10 min), mounted onto microscope glass slides (SuperFrost/Plus, Thermo Scientific), and covered with mounting solution (Roti®-Mount FluorCare, Carl Roth) and a glass coverslip. Digital images of brain sections were acquired sequentially for each fluorophore channel using a confocal laser scanning microscope (Leica TCS SP5, DMI6000 B) with a magnifying objective (10×/0.3 NA, 2× digital zoom) and coupled to a personal computer, running Leica Application Suite Advanced Fluorescence software (LAS AF 2.6). The images were further processed and/or overlaid using ImageJ (version 1.51j8[71]).

## Data analysis

Preprocessing and analysis of the data was performed with custom-written Matlab (R2020a, Mathworks) scripts. Based on the trough-to-peak time (Extended Data Fig. 16), units were classified as putative inhibitory (<0.5 ms) or putative excitatory units (>0.5 ms)[47,72]. Spike counts were binned to 200 ms time frames and the resulting traces were averaged across trials for each recorded condition. To avoid floor effects, units with a baseline firing rate of <0.5 Hz were excluded from the analysis.

To normalize spontaneous activity ($S, S_{ph}$ in Fig. 2), the spontaneous average firing rate was calculated over 3 s after the onset of recording in individual traces during the S condition, and traces with and without photostimulation were divided by this average firing rate. To normalize the evoked responses ($V_{ph}, V_{ph}$ in Fig. 3), the average spiking rate over 1 s before the first visual stimulus was subtracted from each trace. For further normalization of the stimulus-evoked conditions, traces were divided by the first stimulus response amplitude (i.e., 600 ms time interval after the visual stimulus onset) of the control condition ($V$). The resulting normalized traces were then averaged over all units.

To normalize the photostimulation effects, the opto-index ($OI$) was calculated for the $S_{ph}$ trace (Fig. 2):

$$OI = \frac{POST_{Firing} - PRE_{Firing}}{POST_{Firing} + PRE_{Firing}} \tag{1}$$

where $PRE_{Firing}$ represents the firing rate of a unit before photostimulation (average over the 3 s interval after the start of the trial) and $POST_{Firing}$ represents the firing rate during photostimulation (average over the last 3 s interval of the photostimulation interval). Therefore, $OI = 1$ indicates infinite increase in activity from the baseline, $OI = 0$ means no change, while $OI = -1$ indicates complete suppression of the activity.

To quantify the magnitude of visually evoked responses, the difference between response amplitude (average firing within 600 ms after stimulus onset) and the baseline (average firing within 1 s before stimulus onset) was calculated based on each unit's normalized (Fig. 3e) or raw firing rate (Fig. 3f).

For the linear regression (Fig. 3f), the following equation was used:

$$r_{post} = \beta_1 + \beta_2 \cdot r_{pre} + \beta_3 \cdot ph + \beta_4 \cdot ph \cdot r_{pre} \tag{2}$$

Where $r_{pre}$ and $r_{post}$ are the magnitude firing rates before and during photostimulation, respectively, normalized to the unit with the highest, $ph$ represents the presence of photostimulation and is 0 for the V condition and 1 for the V$_{ph}$ condition and $\beta_{1-4}$ are the fitting coefficients of the linear regression model. Therefore, $\beta_3$ and $\beta_4$ represent the effect of photostimulation on the intercept and the slope of the control, respectively, and can point out to a subtractive or divisive effect, respectively.

Putative monosynaptic connections between units were identified based on a fast and transient increase (excitatory connection) or

decrease (inhibitory connection), occurring in the spike cross-correlogram of 2 units with ~2 ms delay[73,74].

For local field potential analysis, the raw signal was downsampled at 1 kHz and low-pass filtered at 300 Hz. Signals of all channels were processed using a fast Fourier transform (FFT) of the photostimulated time interval, averaged over all trials and channels for each recording, and then normalized to their peak value. The spectrogram was calculated by computing the short-time Fourier transform of the signal. The normalized single-sided amplitude spectrum and the power/frequency were quantified by averaging over different frequency bands (slow waves: 0–1 Hz, delta: 1–4 Hz, theta: 4–8 Hz, alpha: 8–12 Hz, beta: 12–30 Hz, gamma: 30–70 Hz).

## Spiking network model

The spiking neural network model consisted of conductance-based excitatory and inhibitory neurons. It was based on the network developed by Sadeh et al.[36] with two changes. Simple integrate-and-fire neurons were used instead of exponential integrate-and-fire neurons and the size of the network was reduced by a factor of 10. While our physiological experiments show indications of putative non-linear effects of lateral interactions[75–79], these were not explicitly implemented in our model to keep the parameter ensemble as simple as possible. This may underestimate the strength of 5-HT$_{2A}$-mediated synaptic effects when modeling its separate experimental activation in local subpopulations[80].

In the neuron model, the membrane potential $V_m$ was described by the following first-order differential equation:

$$C\frac{dV_m}{dt} = -g_L(V_m - E_L) - G_e(t)(V_m - E_e) - G_i(t)(V_m - E_i) \quad (3)$$

where $C$ is the membrane capacitance, $g_L$ is the leak conductance and $E_L$ is the resting potential. When the membrane potential $V_m$ reached $E_T$, a spike was generated and the membrane potential was reset to $V_r$.

The model neurons were driven via excitatory and inhibitory synaptic inputs represented in the second and third terms, respectively. In these terms $E_e$ is the reversal potential of excitatory and $E_i$ is the reversal potential of inhibitory conductances. $G_e(t)$ and $G_i(t)$ are the total excitatory and inhibitory conductances at time $t$, respectively, defined as:

$$G_e(t) = \sum_j g_e(t - t_j) \quad (4)$$

$$G_i(t) = \sum_k g_i(t - t_k) \quad (5)$$

where $t_j$ and $t_k$ are the times when excitatory and inhibitory synaptic events reach a particular post-synaptic neuron, respectively. The conductances $g_e$ and $g_i$ map the timing of synaptic events onto changes in the membrane conductance based on the following

$$g_e(t) = \frac{1}{\tau_e} J_e H(t) \exp\left(1 - \frac{t}{\tau_e}\right) t \quad (6)$$

$$g_i(t) = \frac{1}{\tau_i} J_i H(t) \exp\left(1 - \frac{t}{\tau_i}\right) t \quad (7)$$

$H(t)$ is the Heaviside step function

$$H(t) = \begin{cases} 0 & t \le 0 \\ 1 & t > 0 \end{cases} \quad (8)$$

$J_e$ denotes the peak excitatory conductance and $J_i$ denotes the peak inhibitory conductance. $\tau_e$ and $\tau_i$ are the synaptic time constants of excitatory and inhibitory conductances, respectively.

The network model consists of excitatory (e) and inhibitory (i) subpopulations of size $N_e = 160$ and $N_i = 40$, respectively, that are randomly connected using a binomial distribution with a mean connection probability of $CP$. The peak conductance of each connection is random and drawn from a Gaussian distribution with the mean $J$ and the standard deviation $J/5$. To be consistent with Dale's law, any negative weights were set to zero.

The neurons received background excitatory and inhibitory inputs from independent homogeneous Poisson spike trains firing at rate $\nu_{bkg(e,i)}$, included in the sum in Eq. 4 (for excitation) and Eq. 5 (for inhibition). The effect of 5HT$_{2A}$ activation was modeled via additional background excitatory and inhibitory input firing at rates $\nu_{pert}$ (see Table 2 for parameter values) to different fractions of neurons (note that the parameter values of the fraction of activated units and perturbation strength are in principle interchangeable (Extended Data Fig. 12)). These background inputs are independent homogeneous Poisson spike trains. To account for the different impact of the 5-HT$_{2A}$ receptor-dependent on its localization in pyramidal vs inhibitory neurons (dendrite vs. soma, respectively), the background input was estimated as twice as high in inhibitory units in comparison to excitatory units[81,82]. We modeled the stimulus-evoked responses of neurons by applying a 500-ms long homogeneous Poisson excitatory spike train firing at $\nu_{stim}$ as excitatory inputs to all neurons in both populations. Aggregate results were obtained by simulating 10 different instances of the model (different initializations of the connectivity matrix) and each model 10 times with different realizations of the inputs. The outcomes were then averaged across the 100 simulations.

**Table 2 | Network parameters used in neural network simulations**

| Parameter | Symbol | Value |
|---|---|---|
| Probability of outgoing connections from excitatory neurons | $CP_e$ | 0.15 |
| Probability of outgoing connections from inhibitory neurons | $CP_i$ | 1 |
| Mean peak excitatory conductance | $J_e$ | 1 nS |
| Mean peak inhibitory conductance | $J_i$ | 3 nS |
| Firing rate of background input | $\nu_{bkg_e}$, $\nu_{bkg_i}$ | 970 Hz, 220 Hz |
| Firing rate of the additional background input mimicking systemic activation of 5HT$_{2A}$ receptors | $\nu_{pert}(e)$, $\nu_{pert}(i)$ | 29.5 kHz, 27 kHz |
| Firing rate of the additional background input to excitatory neurons mimicking selective activation of 5HT$_{2A}$ receptors on excitatory neurons | $\nu_{pert}(e \rightarrow e)$, $\nu_{pert}(i \rightarrow e)$ | 50 Hz, 0 Hz |
| Firing rate of the additional background input to inhibitory neuron mimicking selective activation of 5HT$_{2A}$ receptors on inhibitory neurons | $\nu_{pert}(e \rightarrow i)$, $\nu_{pert}(i \rightarrow i)$ | 100 Hz, 0 Hz |
| Firing rate of visual input mimicking the visual stimulus | $\nu_{stim}$ | 200 Hz |

## Statistical analysis

Statistical analysis was performed using Matlab. Two-sided paired sample t-test (with Bonferroni correction for multiple comparisons in Fig. 4a), two-sided one-sample t-test (with Bonferroni correction for multiple comparisons in Figs. 3e and 4b), two-sample Kolmogorov–Smirnov test (Figs. 2c, h and 3c, d), two-sided Mann–Whitney U-test (Fig. 1i), one-way ANOVA (Extended Data Figs. 1 and 2) were used to assess the significance of the values (mean ± SEM).

## Reporting summary

Further information on research design is available in the Nature Portfolio Reporting Summary linked to this article.

## Data availability

The data supporting the findings of this study are available within the main text, the Supplementary Information file or from the corresponding author upon request. Source data are provided with this paper.

## Code availability

The code underlying the modeling is provided as open-source at https://github.com/sencheng/Serotonergic-Modulation[83].

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

## Acknowledgements

We thank Stefan Dobers and Henning Knoop and the RUB mechanical shop for technical support, Caroline Naber for assistance with performing GsX-assays, and Callum White for proofreading the manuscript. This work was supported by Deutsche Forschungsgemeinschaft (DFG) grants: Project ID 122679504 - SFB 874, S.C., S.H., D.J.; Project ID 316803389 - SFB 1280, K.S. and S.H. (Subproject A07), M.D.M. (Subproject A21); JA 945/5-1, D.J.; HE 2471/21-1, S.H.; MA 5806/2-1, M.D.M.; MA 5806/1-2, M.D.M., Project number 492434978 - GRK 2862/1, Subprojects (01, S.H.; 05, M.D.M.; 07, K.S.; 09, I.S.; 10, D.J.); BMBF, ERA-Net Neuron "Horizon 2020", 01EW2104B, D.J.

## Author contributions

Conceptualization: R.B., Z.A., S.H., and D.J. Data curation: R.B. and D.J. Formal analysis: R.B. and D.J. Electrophysiological data acquisition: R.B. and B.B. Electrophysiological data visualization: R.B. and D.J. Neural model: M.M.N. and S.C. 2-Photon data acquisition, GsX assay, analysis, and visualization: L.R. and I.S. Construction and validation of AAV-virus: T.S., H.B., D.E., M.G., and K.S. Histology: R.B., B.B., K.S., and S.T.S. Software: R.B. and Z.A. Supervision: M.D.M, S.H., and D.J. Funding acquisition and resources: M.D.M., S.H., and D.J. Writing—original draft: R.B. and D.J. Writing—review & editing: R.B., S.C., and D.J.

## Funding

## Competing interests

The authors declare no competing interests.
