## [Peer Review File · Nature Communications]

REVIEWER COMMENTS

Reviewer #1 (Remarks to the Author):

Barzan et al investigate the mechanisms of gain control of sensory inputs by serotonin modulation, by stimulating serotonin receptors specifically in excitatory (Exc) and inhibitory (Inh) neurons, and by combining their experimental measurements with computational modelling.

They seek to answer two main questions:

- First, how the effect of such modulation can be inhibitory, while the receptor pathways are excitatory in single cells. Their answer to this question is the disynaptic interaction of exc-inh in the local circuit.
- Second, how this suppressive effect can spare the baseline activity, and only affect the evoked responses. They suggest that this can only be explained if the network is operating in fluctuation-driven regimes of activity (which typically emerges as a result of balance of exc-inh in recurrent networks).

While the first result is rather expected, the second is interesting. The problem is that the main evidence to support this is coming from the computational modelling, and there's no experimental evidence provided to support that (either directly, or indirectly via the predictions).

Major comments:

1) More work is needed to reveal the exact mechanism. The insights obtained from computational modelling (Fig. 4) are interesting. But in its current form the study falls short of convincing us that the proposed mechanism is indeed responsible for the results in biological networks. Ideally this should be seen in subthreshold data. Knowing that this can technically be challenging, would there be any specific predictions from the model that can be tested by the experiments feasible with the employed techniques? It would be specifically good to dissociate the suggested mechanism from alternative mechanisms. For instance, wouldn't a network operating in a mean-driven regime be able to behave the same, if a general inhibition increased the distance to threshold (and hence led to divisive inhibition)? Or even a single-cell mechanism like shunting inhibition. A more convincing argument for the biological relevance of the suggested mechanism is therefore needed.

2) Some of the analyses need further work. Better comparisons and more careful controls are needed (between experiments, but also between experimental and modelling results; see below for specific comments).

3) Regarding the modelling results (Fig. 4): why not repeating the same procedure as the experiments (especially the evoked protocol in Fig. 3)? Visual stimulation can be delivered in a similar manner and compared to experimental results (especially, cf. Fig. 3b). It would be good to show more directly if we get the divisive effect in a similar way in the model too (cf. Fig. 3d). This can potentially provide further predictions or specific metrics to be tested with experiments (re point 1).

4) Writing needs improvement (especially Abstract, Introduction, and Results sections; see specific examples below).

Specific comments:

Regarding analysis in Fig. 2:

L50-52: “led to an increase in spontaneous activity of both excitatory (Fig. 2a, dark green panel, “direct effect”, +48%) and inhibitory neurons (Fig. 2a, light red panel, “indirect, polysynaptic effect”, +70%).”

How is this quantified? From the figure, it seems that the increase for exc neurons (Fig. 2a, left) is more than (or at least comparable to) the increase for inhibitory neurons (Fig. 2a, right).

In Fig. 2b:

There is a ramping increase in inhibition (Fig. 2b, left panel). Shouldn't we expect to see a ramping suppression (downwards), as a result of disynaptic interaction, in exc neurons (Fig. 2b, right)? It is confusing that the indirect effect looks completely flat.

It has been argued that a wide distribution of OI effects is observed for inh neurons in (b) – and hence analysis is broken down to OI+ and OI- neurons. A similar wide distribution of effects seems to also exist for exc (and inh) neurons in (a) – as evident from comparing OI distributions in c and d (last panels). Would we learn something from performing the same analysis for neurons in (a)?

OI index:

The way the index is defined (Eq. 1) should make it very sensitive to baseline firing rates – specifically amplifying the effect for neurons with low baseline firing rates. The results should be controlled for this. This can be done either by developing some other measure(s), or/and by performing further control analyses (e.g., how OI is correlated with the firing rates of units, and performing the analysis for different percentiles and testing if the results hold).

Distributions of OIs in panel c (last column) are difficult to read / compare – better visualization is needed.

Regarding analysis in Fig.3:

Don't we have a distribution of effects here too, similar to what shown in Fig. 2?

The results are now shown for the average effects only, is this a uniform effect across all neurons? If not, it would be good to have some further analysis and illustration of the distributions (which can also be useful for shedding more light on the mechanism).

*

Writing needs improvement. Some examples below:

From Abstract:

Rewrite:

“Combining these results within a realistic cortical network model **reveals a conductance-driven polysynaptic mechanism that provides control of stimulus-evoked gain** without affecting baseline levels of ongoing activity.”

“Our study opens new avenues for **scrutinizing neuromodulation by GPCRs and their impact on sensory input and ongoing brain broadcasts**.”

From Introduction:

Rewrite:

L1: “One prominent function of modulatory systems on cortical processing is their influence on ...”

L19: Contribution of .. in gain modulation; in -> to

L26: “Hence, in this study we take the 5-HT_{2A} receptor as ****an exemplar of isolating a specific coherent function of a modulatory system**** ...”

Hard to understand how a receptor can be an exemplar of a function.

From Results:

L77: “****Activation**** of 5-HT_{2A} in PV neurons ****participates**** in controlling visual gain in excitatory neurons”

Hard to understand.

L103: “To substantiate our results, we employed a spiking cortical network model ...”

Modelling is to understand, not to substantiate (especially given the gap between the experimental and modelling results here).

The way it’s referred to different subtypes is confusing in several instances:

L14: “different classes of cortical neurons (pyramidal, parvalbumin (PV), ...”

L36: “expressed in different classes of cortical neurons (pyramidal, PV, see Fig. 1c)”

Miscellaneous:

L57: Fig. 2c referred before Fig. 2b

Table 2: the subindices for the ν_{pert} parameters are confusing

The code for reproducing the results of computational modelling is not provided.

Reviewer #2 (Remarks to the Author):

This manuscript addresses a significant topic within the subject of neural circuit modulation by serotonin-2A (5-HT_{2A}) receptors. The authors note that 5-HT_{2A} activation in V1 causes global

decreases in visual response gain with little effect on baseline firing rates despite 5-HT2A activation having excitatory effects in neurons that express the receptor. The system is further complicated by expression of 5-HT2A in distinct classes of neurons with different output types (excitatory and inhibitory). The authors address these observations by exogenously expressing an optogenetic GPCR in genetically defined cell types and assessing the outcomes of activating the receptor on spontaneous and visually evoked activity with electrophysiology. They observe that 5-HT2A activation specifically in interneurons causes bidirectional effects on firing rates across the population, whereas activation in excitatory neurons causes only enhanced firing. Finally, they provide a model that supports a polysynaptic mechanism for 5-HT2A-based modulation of gain control that leaves baseline activity unaffected.

As written, there seems to be a significant disconnect between the narrative and the actual experiments - after reading the title and introduction one might expect that the authors engineered a tool to optogenetically activate 5-HT2A receptors in neurons that endogenously express the protein. However, it appears instead that this work is more broadly testing a theory that GPCR signaling in a subset of neurons can widely alter conductances in the network to change response gains, with a tool that does not recapitulate the normal distribution or action of 5-HT2A receptors. While interesting and useful from a theoretical standpoint, the overall framing of the work seems misleading, and the ultimate connection between the experiments and modeling performed here and endogenous 5-HT2A activation in vivo is unclear.

Major comments:

-The critical tool for these experiments is the “light-activated 5-HT2A receptor,” however no substantial description or characterization of this receptor seems to be provided. Given no previous study is cited for this tool, its presumed development in this study supports the novelty of the work, but more information about exactly what the protein is and how it works is critical for the reader. In the methods, the construct is defined as pAAV-CMV-floxed-mOpn4L-eGFP-5-HT2ACT. This would suggest the authors are using a melanopsin receptor linked to the C-terminus of the human 5-HT2A receptor, presumably so that the Gq-coupled melanopsin receptor will be trafficked within the cell as a 5-HT2A receptor would. If this is true, the wording throughout the manuscript should clarify that the authors are not activating endogenous 5-HT2A receptors (e.g. line 34, “...first time optogenetic control of an endogenous receptor type” and line 37, “light-activated 5-HT2A” among many other places throughout the text), and instead are using light-activated melanopsin to achieve Gq-signaling to mimic 5-HT2A activation. If I have misinterpreted the technique, the authors clearly need to better explain how this is a light-activated 5-HT2A receptor.

-Cre-dependent expression of a Gq-coupled receptor in excitatory and inhibitory populations is a useful experimental manipulation, however there is no discussion in the text of how this relates to endogenous 5-HT2A expression. Only subsets of excitatory and inhibitory neurons express 5-HT2A

in V1, so the authors are presumably expressing this construct in neurons that do not endogenously express 5-HT2A. It would be useful if the authors could clarify how these experiments relate to the endogenous system (that has layer and cell-type specificity with excitatory and inhibitory classes), and whether conclusions can really be made specifically about how 5-HT2A activation affects cortical microcircuits or whether conclusions more broadly about GPCR signaling are more appropriate.

-The modeling results that increased conductance due to widespread 5-HT2A activation can lead to decreased visual response gain are intuitive, however the fact that activation in 50% of neurons results in gain decreases of about 35% seems inconsistent with in vivo data, where decreases of more than 50% in visual response gain result with endogenous 5-HT2A expression in much less than half of the V1 population. Data across the range of units activated seem to suggest the model only explains a portion of the change in response gain seen biologically. This is in contrast to the Discussion where the results are described as “similar to values in the literature” (Line 207-8), but that’s only true if the majority of cortical neurons express 5-HT2A which is not supported by previous evidence. Thus, the authors should address why the model may not account for the full effect.

Minor comments:

-Given that it’s not possible to know whether a recorded neuron is expressing the viral construct, it would be useful for the authors to note this in the text explicitly, and that the recorded effects on a given neuron could be due to direct and/or indirect (polysynaptic) mechanisms.

-It would be helpful to know the quantitative metrics that were used to separate recorded units into putative excitatory and inhibitory, with some reference to previous publications that use those metrics.

-The authors should clarify whether the recordings were performed in awake or anesthetized animals - the latter is assumed based on the methods, but it’s not explicitly stated. Assuming they’re anesthetized, the authors should discuss how this could affect the interpretation of their results in the context of other findings that have largely been in awake animals - cortical dynamics are very different under anesthesia, and inhibition is particularly affected.

-Line 69: “In sharp contrast to the overall activating effect across the pool of inhibitory neurons” should this be “excitatory” neurons?

-Figure 1A: consider labeling pyramidal and PV neurons.

-Figure 1B legend: “without affecting baseline levels of activity.”

-Figure 2A/B: The color scheme becomes confusing here. Labeling the columns as recorded excitatory/inhibitory and rows as activated excitatory/inhibitory (which requires switching the graphs in B), or something similar, could provide more clarity.

-Figure 2B: The OI+ and OI- traces that were added to the left panel should also be added to the other panels as well (including in A) or removed.

-Figure 2D: The OI plot and histogram nicely show the bidirectional change for inh and largely unidirectional for exc, but in as with the above comment the bar plot on the left should only have the OI+ and OI- broken out for both the inh and exc data or should not be done at all. Also, swapping the order of the exc/inh data in the OI plot to match the one above would be more intuitive

-Figure 3: The authors should again consider changing the order of the 4x4 plots in A/B as suggested in Figure 2 comments.

-Figure 4: It would be useful to see how varying the %units with activated 5-HT_{2A} affect firing rates and changes in magnitude for the “all neurons” condition. For example, a plot with x-axis %excitatory, y-axis %inhibitory, and colormap for firing rate or change in magnitude. This would help the reader understand how the effects of 5-HT_{2A} activation in the two populations interact.

-The authors might consider alternative color schemes rather than red/green to help color-blind readers more easily interpret the figures.

Reviewer #3 (Remarks to the Author):

In this manuscript, Barzan and colleagues report the effects of activating the 5-HT_{2A} receptor on visual cortical activity in mice. Receptor activation was achieved via a novel construct that

supposedly allowed for optogenetic control of the receptor. They then proceeded to show effect of the manipulation on visually evoked activity.

The study is challenging to understand because key controls are missing. One, there is hardly any calibration and validation of the mOpn4L-5HT2A that allows them to manipulate the receptor signaling. Does the construct work as intended? How do we know it cause 5-HT2A-like function in vivo? Where is the mutant protein localized in the cell? None of this was presented quantitatively. Second, the characterization of visual responses is very simplistic. Much more can be tested by varying the contrast of the stimuli or the orientation tuning. It is not clear if the manipulation changes contrast response or orientation tuning, because the authors tested only a single grating at a single contrast, which severely limit the evidence supporting their claim for whether the suppression was divisive or subtractive.

As these two points are foundational to understand the study, but are unfortunately missing, it is not obvious that the authors have done what they set out to do.

Major comments:

- Control data is missing to show that photo-stimulation of mOpn4L-5-HT2A activates the G-protein signals in the targeted cell types. It is also not clear where this construct is localized in the cell. More importantly, there is no data showing that activation of this construct, actually drives downstream signaling in comparable manner as normal 5-HT2A receptors. Put in other words, what is the difference between the current approach with other methods like Gq-DREADD which also targets the Gq signaling?

- Figure 2b, separate depiction of SphOI+ and SphOI- groups do not seem justified as these are not distinct groups (looking at histogram in Fig. 2d). It's a continuous spectrum of responses. As the author stated in line 62 "a distribution of the OI that is symmetrically centered around 0". This then carry over to Fig. 2d, where now statistical significance was separately determined for SphOI+ and SphOI- group - of course these are large because the authors arbitrarily divided the group based on the response and then tested the significance of such responses.

- The way the visual responses were mapped is rudimentary. Much more can be tested by varying the contrast of the stimuli or the orientation tuning. It is not clear if the manipulation changes contrast response or orientation tuning. Here the authors show what they say is a divisive suppression, but that's divisive across the population of cells recorded. If they had varied the contrast, then we can see if it is divisive for single neurons, or whether some neurons may show divisive suppression while others have subtractive suppression.

Minor comments:

- Fig. 1d - what is the fidelity of separation into regular and fast-spiking populations? The authors should provide a histogram so we can see if the distribution of the waveform features used to make the split.

- Fig. 1d - you can say these populations are fast-spiking and regular-spiking cells, but we do not know if these are true excitatory and inhibitory neurons

- What is the specificity for pyramidal neuron targeting using the NEX-Cre mice? That is, what fraction of transgene-expressing cells are not pyramidal neurons, and what fraction of pyramidal neurons did not express the transgene? This is not quantified.

- Line 49 - provide more specifics about the optogenetic activation (pulse duration, frequency, etc.)

- Figure 2c and d, left panels, what are those colored bars at the top of the plots?

- Figure 2c and d, the right panel's statistical test was specified as paired t-test. What about the left and middle panels?

- It's recommended to avoid using red and green as the colors because they are not easily distinguished for people who are colorblind.

- The description of 'polysynaptically transmitted hyperpolarization via PV neurons in which the 5-HT_{2A} receptor was photoactivated.' was confusing and it is not clear what the authors mean (line 74/75)

- There is soma labeling but it is thought that much of the endogenous 5-HT_{2A} receptors are localized to apical dendrites.

- It is not appropriate to use a single orientation to measure the visual responses, as V1 neurons exhibit different tuning so single-orientation will drive few cells well, but also many cells suboptimally, affecting how one may consider the 'average response'.

- Fig. 3 includes result of pyramidal neuron stimulation, but the title only mentioned stimulation of parvalbumin interneurons

Point-by-point response to the reviewer comments

We thank the reviewers very much for their helpful and constructive comments on our manuscript. We have performed new experiments, where we used 2-photon imaging (in brain slices of V1) in combination with pharmacology in order to verify the specificity of the used optogenetic tool. Moreover, motivated by the advice that the experimental data and the model need a better connection, we performed a series of new *in vivo* experiments and new analysis that directly tested the predicted model effects on joint 5-HT_{2A} signaling in inhibitory and excitatory neurons.

Please see below our point-by-point reply to the reviewer comments and the revised version of our manuscript with all changes marked.

Reviewer #1 (Remarks to the Author):

Barzan et al investigate the mechanisms of gain control of sensory inputs by serotonin modulation, by stimulating serotonin receptors specifically in excitatory (Exc) and inhibitory (Inh) neurons, and by combining their experimental measurements with computational modelling.

They seek to answer two main questions:

- First, how the effect of such modulation can be inhibitory, while the receptor pathways are excitatory in single cells. Their answer to this question is the disynaptic interaction of exc-inh in the local circuit.
- Second, how this suppressive effect can spare the baseline activity, and only affect the evoked responses. They suggest that this can only be explained if the network is operating in fluctuation-driven regimes of activity (which typically emerges as a result of balance of exc-inh in recurrent networks).

While the first result is rather expected, the second is interesting. The problem is that the main evidence to support this is coming from the computational modelling, and there's no experimental evidence provided to support that (either directly, or indirectly via the predictions).

We performed new *in vivo* experiments, where we used optrode electrophysiology and activated the 5-HT_{2A} receptor pathway in both in excitatory and inhibitory (Exc/Inh) neurons simultaneously. The new data supports our main hypothesis and computational model that systemic activation of the 5-HT_{2A} pathway stabilizes baseline activity levels while selectively controlling the gain of visual response.

Major comments:

1) More work is needed to reveal the exact mechanism. The insights obtained from computational modelling (Fig. 4) are interesting. But in its current form the study falls short of convincing us that the proposed mechanism is indeed responsible for the results in biological networks. Ideally this should be seen in subthreshold data. Knowing that this can technically be challenging, would there be any specific predictions from the model that can be tested by the experiments feasible with the employed techniques?

Comparison between modeling and experimental results expanded:

i.) New Fig. 4 and Extended Data Fig. 7: We agree with the reviewer that specifically subthreshold recordings would be desirable but would come with additional technical challenges, which would also go beyond our expertise. We therefore decided to directly test the model predictions with new, however, still challenging, experiments. The reviewer may share our enthusiasm that these experiments indeed support the model predictions. Specifically, the new experiments show that activating 5-HT_{2A} simultaneously in both pyramidal and PV neurons does not affect baseline and reduces the visual gain in both subpopulations, a result that previously was predicted by systemic 5-HT_{2A} activation in our model.

ii.) New Fig. 2c-e/h-j and new Extended Data Fig. 8: Additionally, we performed further model analyses and found that when 5-HT_{2A} is activated in the Inh-subpopulation (dark blue in new Extended Data Fig. 8b), the directly stimulated units show increase in opto-index (OI), while indirectly affected Inh-units show a decrease in OI. In contrast, the Exc-subpopulation, which is only indirectly stimulated, maintains a fairly compact OI distribution. This prediction matches the experimental data (new Fig. 2h) where the shape of the OI distribution of the Exc-population remains roughly similar to the control condition while the distribution of the Inh-population becomes much broader following photostimulation.

iii.) Analog (but less pronounced) patterns of widening of the OI distributions emerge when 5-HT_{2A} is activated in the Exc-subpopulation (compare new figures Extended Data Fig. 8a and Fig. 2c).

iv.) Note that the widening of the OI distributions during baseline firing only occurs if an intermediate fraction of the subpopulation is affected by 5-HT_{2A} activation (Extended Data Fig. 8). Interestingly, also the strength in gain suppression of visual input is in the range of the biological data when an intermediate fraction of the subpopulation is affected by 5-HT_{2A} activation (cf. Fig. 5d).

It would be specifically good to dissociate the suggested mechanism from alternative mechanisms. For instance, wouldn't a network operating in a mean-driven regime be able to behave the same, if a general inhibition increased the distance to threshold (and hence led to divisive inhibition)? Or even a single-cell mechanism like shunting inhibition. A more convincing argument for the biological relevance of the suggested mechanism is therefore needed.

We dissociated the suggested fluctuation-driven from alternative mechanisms (please see also the respective added text in the manuscript with tracked changes).

i.) New Fig. 5c shows modeling results after activating 5-HT_{2A} in both subpopulations in the mean-driven regime. The effects were roughly additive, i.e., the sum of activating 5-HT_{2A} in each subpopulation separately (cf. Fig. 5a and b). Thus, in this case, systemic activation yielded negligible changes in baseline firing and only little gain modulation in Exc-units, while the baseline activity in the Inh-population increased without effecting response gain. Hence, this outcome is different from our experimental observations (new Fig. 4) and the model's behavior in the fluctuation-driven regime (Fig. 5d) – and also different from our observations in a previous study (Azimi et al., 2020).

We added (p7): “When modeling the above effects in a mean driven regime, keeping all parameters constant, the results were roughly additive, yielding the sum of activating 5 HT_{2A} in each subpopulation (cf. Fig. 5a and b) separately. That is, even though excitatory units displayed negligible changes in baseline firing as in the experiments, a significant modulation of evoked response gain was absent in these units (Fig. 5c, triangles). In addition, contrary to the systemic activation in the experiments, the baseline activity in the inhibitory population increased without any effect on response gain (Fig. 5c, circles). Hence, for the mean-driven

the outcome was dramatically different from the experimental observations (Fig. 4) and therefore, the simple linear model substantially fails in explaining the experimentally revealed values.”

ii.) New Extended Data Fig. 9: Changing the stimulation parameters would not change the results qualitatively. Since the Exc-units exhibited opposite responses when 5-HT_{2A} was activated in the two different subpopulations (Fig. 5a, dark orange vs Fig. 5b, light orange and new Extended Data Fig. 9a left vs. Fig. 9b right) these effects could potentially cancel each other and keep the baseline firing rate unchanged. However, by contrast, inhibitory neurons increased their firing rates in both scenarios (Fig. 5a, light blue vs Fig. 5b, dark blue and new Extended Data Fig. 9a right vs. Fig. 9b left). Thus, in a mean-driven systemic activation scenario the Inh-units increase their baseline firing rates and also showed no gain modulation (Fig. 5c right), which is contradictory to the observed biological findings.

iii.) Shunting inhibition, in a general sense, might indeed contribute to the reduced gain after systemic activation of 5-HT_{2A}. However, in a stricter sense and to the best of our knowledge, shunting is referred to as a push-pull-like mechanism following rapid changes in input. To avoid misunderstanding, and to keep or wording neutral in this respect, we tried to avoid this terminus. Moreover, it is still an ongoing debate in how far shunting contributes to divisive (as observed here) rather than to subtractive effects. Clearly, we cannot rule-out that 5-HT_{2A} activation induces single-cell effects in addition to the network effects that we study here.

2) Some of the analyses need further work. Better comparisons and more careful controls are needed (between experiments, but also between experimental and modelling results; see below for specific comments).

Addressed - see our replies to the specific comments.

3) Regarding the modelling results (Fig. 4): why not repeating the same procedure as the experiments (especially the evoked protocol in Fig. 3)? Visual stimulation can be delivered in a similar manner and compared to experimental results (especially, cf. Fig. 3b). It would be good to show more directly if we get the divisive effect in a similar way in the model too (cf. Fig. 3d). This can potentially provide further predictions or specific metrics to be tested with experiments (re point 1).

Please also see the above new analysis. It shows a bimodal distribution of effects in the model (new Extended Data Fig. 8) in line with the experimental observation that activation of 5-HT_{2A} in PV interneurons leads to both facilitation and suppression of spontaneous activity in Inh neurons.

New Extended Data Fig. 10: We further show that the model also predicts a divisive suppression of Exc units upon 5-HT_{2A} activation in inhibitory units.

The model does not implement adaptation effects as observed for the biological data. Thus, each visually evoked peak of activity is identical. To avoid an overly redundant depiction we have chosen a more compact way of presenting the modeling data.

4) Writing needs improvement (especially Abstract, Introduction, and Results sections; see specific examples below).

Done – We improved the given examples below and conducted further proofreading by two colleagues that are English native speakers.

Specific comments:

Regarding analysis in Fig. 2:

L50-52: “led to an increase in spontaneous activity of both excitatory (Fig. 2a, dark green panel, “direct effect”, +48%) and inhibitory neurons (Fig. 2a, light red panel, “indirect, polysynaptic effect”, +70%).”

How is this quantified? From the figure, it seems that the increase for exc neurons (Fig. 2a, left) is more than (or at least comparable to) the increase for inhibitory neurons (Fig. 2a, right).

Both traces (with and without photostimulation) were normalized to the initial baseline activity of the trace without photostimulation, while the bar plot quantified traces normalized to their own individual baseline activity.

We re-calculated the normalized traces (new Fig. 2c-e and h-j) to show normalization to their own baseline activity, as do the bar plots.

In Fig. 2b:

There is a ramping increase in inhibition (Fig. 2b, left panel). Shouldn't we expect to see a ramping suppression (downwards), as a result of disynaptic interaction, in exc neurons (Fig. 2b, right)? It is confusing that the indirect effect looks completely flat.

A possible explanation is that excitation can increase the normalized firing rate from 1 to infinite, whereas suppression can only decrease the normalized firing rate from 1 to 0. This can have an impact on the depiction of the effect, as prolonged excitation can lead to continuously increasing firing rate, whereas inhibition could lead to a floor effect in the firing rate after a shorter time interval. Thus, because of floor effects the traces with negative OIs (new Fig. 2i, blue and orange stippled lines for Inh and Exc neurons, respectively) may prevent a further downward ramping in both cases.

It has been argued that a wide distribution of OI effects is observed for inh neurons in (b) – and hence analysis is broken down to OI+ and OI- neurons. A similar wide distribution of effects seems to also exist for exc (and inh) neurons in (a) – as evident from comparing OI distributions in c and d (last panels). Would we learn something from performing the same analysis for neurons in (a)?

New Fig. 2c-e and h-j – We thank the reviewer for this helpful hint. We performed the suggested analysis, in which our reasoning (see also our reply above) was further confirmed:

Quantification of activity changes shows a clear increase in activity in neurons with OI-positive (Fig. 2e-g, NEX-Cre, exc: +76±13%, inh: +169±59%), while a more modest decrease in OI-negative neurons (NEX-Cre, exc: -19±3%, inh: -20±3%). The same quantification in

experiments in PV-Cre mice reveals a strong excitatory effect in interneurons with only a weak increase in activity of excitatory neurons (Fig. 2h-j, inh: OI+: $83\pm 15\%$, exc, OI+: $+17\pm 6\%$), and a similar inhibitory effect across both populations (inh, OI-: $-39\pm 4\%$, exc, OI: $-41\pm 2\%$). This information was now added in the main text of the manuscript.

Overall, the newly performed analysis supports the effects of 5-HT_{2A} pathway manipulation as previously described: with strong excitation in pyramidal neurons, while activation in PV interneurons causes a mixed effect in inhibitory neurons and mostly an inhibition in excitatory neurons.

OI index:

The way the index is defined (Eq. 1) should make it very sensitive to baseline firing rates – specifically amplifying the effect for neurons with low baseline firing rates. The results should be controlled for this. This can be done either by developing some other measure(s), or/and by performing further control analyses (e.g., how OI is correlated with the firing rates of units, and performing the analysis for different percentiles and testing if the results hold).

In comparison to a regular normalization, which would scale the inhibitory effects between 0 and 1 and the excitatory effects between 1 to $+\infty$, and therefore give more weight to excitatory effects, the opto-index scales the inhibitory effect between -1 to 0 and the excitatory between 0 to 1. Therefore, the opto-index was introduced to balance the weight of inhibitory and excitatory effects. Please also note that units with average firing rate lower than 0.5 Hz were excluded from analysis.

New Extended Data Fig. 4: The suggested analysis, OI vs. firing rate, was performed. The average OI was calculated for each quartile of the firing rate data set. While in few conditions a weak trend was visible, correlation analysis between OI and firing rate was also performed, revealing a squared Pearson correlation coefficient (R^2) between 0.01 and 0.16, therefore showing none to very little correlation.

Distributions of OIs in panel c (last column) are difficult to read / compare – better visualization is needed.

Done – To improve visualization the distributions of OIs are now plotted as violin plots (new Fig. 2c and h).

Regarding analysis in Fig.3:

Don't we have a distribution of effects here too, similar to what shown in Fig. 2?

The results are now shown for the average effects only, is this a uniform effect across all neurons? If not, it would be good to have some further analysis and illustration of the distributions (which can also be useful for shedding more light on the mechanism).

Done – Thanks again for this remark. As stated above, analyzing the distributions helped indeed to further reveal similarities between the experimental and the modeling data. The respective distributions are now shown in new Fig. 3c and d.

Writing needs improvement. Some examples below:

From Abstract:

Rewrite:

“Combining these results within a realistic cortical network model **reveals a conductance-driven polysynaptic mechanism that provides control of stimulus-evoked gain** without affecting baseline levels of ongoing activity.”

Rewritten – however, please note, we not entirely sure whether this has been addressed in full.

“Our study opens new avenues for **scrutinizing neuromodulation by GPCRs and their impact on sensory input and ongoing brain broadcasts**.”

Rewritten

From Introduction:

Rewrite:

L1: “One prominent function of modulatory systems on cortical processing is their influence on ...”

Reformulated

L19: Contribution of .. in gain modulation; in -> to

Done

L26: “Hence, in this study we take the 5-HT2A receptor as **an exemplar of isolating a specific coherent function of a modulatory system** ...”

Hard to understand how a receptor can be an exemplar of a function.

Reformulated

From Results:

L77: “**Activation** of 5-HT2A in PV neurons **participates** in controlling visual gain in excitatory neurons”

Hard to understand.

Rephrased

L103: “To substantiate our results, we employed a spiking cortical network model ...”

Modelling is to understand, not to substantiate (especially given the gap between the experimental and modelling results here).

We skipped the first part of the sentence.

The way it's referred to different subtypes is confusing in several instances:

L14: "different classes of cortical neurons (pyramidal, parvalbumin (PV), ..."

L36: "expressed in different classes of cortical neurons (pyramidal, PV, see Fig. 1c)"

Reformulated - Thank you for these hints.

Miscellaneous:

L57: Fig. 2c referred before Fig. 2b

Corrected.

Table 2: the subindices for the ν_{pert} parameters are confusing

We agree with the reviewer and changed the notation. For instance, instead of $\nu_{\text{pert}_{i \rightarrow i}}$ we now use the notation $\nu_{\text{pert}}(i \rightarrow i)$

The code for reproducing the results of computational modelling is not provided.

Upon publication, the code will be released as open-source on our groups Github account (<https://github.com/sencheng>) like we did for other projects.

Reviewer #2 (Remarks to the Author):

This manuscript addresses a significant topic within the subject of neural circuit modulation by serotonin-2A (5-HT_{2A}) receptors. The authors note that 5-HT_{2A} activation in V1 causes global decreases in visual response gain with little effect on baseline firing rates despite 5-HT_{2A} activation having excitatory effects in neurons that express the receptor. The system is further complicated by expression of 5-HT_{2A} in distinct classes of neurons with different output types (excitatory and inhibitory). The authors address these observations by exogenously expressing an optogenetic GPCR in genetically defined cell types and assessing the outcomes of activating the receptor on spontaneous and visually evoked activity with electrophysiology. They observe that 5-HT_{2A} activation specifically in interneurons causes bidirectional effects on firing rates across the population, whereas activation in excitatory neurons causes only enhanced firing. Finally, they provide a model that supports a polysynaptic mechanism for 5-HT_{2A}-based modulation of gain control that leaves baseline activity unaffected.

As written, there seems to be a significant disconnect between the narrative and the actual experiments - after reading the title and introduction one might expect that the authors engineered a tool to optogenetically activate 5-HT_{2A} receptors in neurons that endogenously express the protein. However, it appears instead that this work is more broadly testing a theory that GPCR signaling in a subset of neurons can widely alter conductances in the network to change response gains, with a tool that does not recapitulate the normal distribution or action of 5-HT_{2A} receptors. While interesting and useful from a theoretical standpoint, the overall framing of the work seems misleading, and the ultimate connection between the experiments and modeling performed here and endogenous 5-HT_{2A} activation in vivo is unclear.

We thank the reviewer for raising this important point. To avoid confusion, we now clarified the properties of the used tool from the beginning.

In the introduction we wrote: “Importantly, the C-terminus of the used construct warrants its expression in the endogenous receptor-specific cellular domains (Eickelbeck et al., 2019), providing a light-activatable functional equivalent to endogenous 5-HT_{2A} receptor signals. Moreover, targeting GPCR tools to receptor-specific domains triggers downstream kinetics with similar strength as compared to their endogenous analogues and ensures no overshoot when comparing light-activated to the native cellular effects (Oh et al., 2010).”

In the discussion we added (p9): “Also, the expression pattern of our construct does not concur with the normal complex distribution of 5-HT_{2A} receptors across cortical layers (Weber and Andrade, 2010), which naturally serves further signal tuning within a spectrum of functions.”

Major comments:

-The critical tool for these experiments is the “light-activated 5-HT_{2A} receptor,” however no substantial description or characterization of this receptor seems to be provided. Given no previous study is cited for this tool, its presumed development in this study supports the novelty of the work, but more information about exactly what the protein is and how it works is critical for the reader. In the methods, the construct is defined as pAAV-CMV-floxed-mOpn4L-eGFP-5-HT_{2A}ACT. This would suggest the authors are using a melanopsin receptor linked to the C-terminus of the human 5-HT_{2A} receptor, presumably so that the G_q-coupled melanopsin receptor will be trafficked within the cell as a 5-HT_{2A} receptor would.

As addressed above and below. As a further comment, in a previous study of our group, a chimeric version of the current construct with a different fluorophore was used. It did not permit cell type-specific transfection but otherwise mimicked 5-HT_{2A} receptor signaling (see ref. 20, Eickelbeck et al., 2019 (previously ref. 27). However, as the construct in the present study was different and the former did not allow for a conclusive result regarding baseline analysis, we were reluctant to additionally cite it in the method section.

New Fig. 1g-I and Extended Data Fig. 1: Importantly, we now show directly in V1 brain slices that our construct activates specifically the G_q-coupled pathway.

If this is true, the wording throughout the manuscript should clarify that the authors are not activating endogenous 5-HT_{2A} receptors (e.g. line 34, “...first time optogenetic control of an endogenous receptor type” and line 37, “light-activated 5-HT_{2A}” among many other places throughout the text), and instead are using light-activated melanopsin to achieve G_q-signaling to mimic 5-HT_{2A} activation. If I have misinterpreted the technique, the authors clearly need to better explain how this is a light-activated 5-HT_{2A} receptor.

Clarified in parts above – In addition, throughout the manuscript, we amended the wording to avoid misunderstanding. Line 34, “...first time optogenetic control of an endogenous receptor type” now reads (p3): “To the best of our knowledge, this is the first time an optogenetic control of a single endogenous receptor pathway (5-HT_{2A}) specifically activated in two different types of cortical neurons (i.e., pyramidal and PV neurons in the visual cortex; see Fig. 1c-f) to explore basic polysynaptic mechanisms of gain control by GPCR signaling has been used.” Line 37, “light-activated 5-HT_{2A}” changed to (p3): “To achieve this, we expressed a chimeric construct consisting of light-activated mouse melanopsin (mOpn4L, targeted into 5-HT_{2A} receptor domains³¹) to trigger G_q-signaling (Fig. 1g-i; Extended Data Fig. 1) that mimics 5-HT_{2A} receptor activation ...”

-Cre-dependent expression of a G_q-coupled receptor in excitatory and inhibitory populations is a useful experimental manipulation, however there is no discussion in the text of how this relates to endogenous 5-HT_{2A} expression. Only subsets of excitatory and inhibitory neurons express 5-HT_{2A} in V1, so the authors are presumably expressing this construct in neurons that do not endogenously express 5-HT_{2A}. It would be useful if the authors could clarify how these experiments relate to the endogenous system (that has layer and cell-type specificity with excitatory and inhibitory classes), and whether conclusions can really be made specifically about how 5-HT_{2A} activation affects cortical microcircuits or whether conclusions more broadly about GPCR signaling are more appropriate.

We further clarified this point and added in the discussion (p9):

“Thus, the detailed picture of the underlying mechanisms in terms of layer-specific circuitries needs further study. Moreover, expression strength of optogenetic tools can vary, as also accounted for in our model (Extended Data Fig. 11).” ... (p9-10): “However, while the value of effect sizes can deviate between the light-activated and the native 5-HT_{2A} system, it appears unlikely that our manipulation imposes fundamental different operations across the native functional architecture. Our major point that the joint activation of the 5-HT_{2A} pathway (i.e. simultaneous G_q signaling in pyramidal and PV neurons) in a balanced regime leads to a robust qualitative change in the cortical state allowing for gain control of external input without changes in baseline activity is further supported by previous data in the native system (Azimi et al., 2020).”

-The modeling results that increased conductance due to widespread 5-HT_{2A} activation can lead to decreased visual response gain are intuitive, however the fact that activation in 50% of neurons results in gain decreases of about 35% seems inconsistent with in vivo data, where decreases of more than 50% in visual response gain result with endogenous 5-HT_{2A} expression in much less than half of the V1 population. Data across the range of units activated seem to suggest the model only explains a portion of the change in response gain seen biologically. This is in contrast to the Discussion where the results are described as “similar to values in the literature” (Line 207-8), but that’s only true if the majority of cortical neurons express 5-HT_{2A} which is not supported by previous evidence. Thus, the authors should address why the model may not account for the full effect.

There may have been a misunderstanding here. In Fig. 5d (previously 4c), the region shaded in blue indicates the range of gain modulation that is consistent with current available data from different experiments. To achieve these values of gain modulation in our model, the 5-HT_{2A} pathway has to be activated in somewhere between 30%-80% of neurons. If gain modulations are at the lower end, G_q activation in a minority of neurons (30%) is sufficient to account for

the entire effect for the given reference and our new data. Indeed, higher values of gain modulation are obtained, if the 5-HT_{2A} pathway is activated in a larger fraction of neurons, possibly overestimating endogenous percentages. However, please note that our model simulations yield similar results (i.e., changes in firing rate) for different parameter values of percentage activated units and perturbation strength (new Extended Data Fig. 9). Therefore, given that excitation and inhibition is balanced, the precise values of parameters seem less important for the qualitative result that gain changes occur despite a lack of changes in baseline when the 5-HT_{2A} pathway is activated systemically, and that this pattern is completely different if 5-HT_{2A} activation is targeted to only one subpopulation.

Minor comments:

-Given that it's not possible to know whether a recorded neuron is expressing the viral construct, it would be useful for the authors to note this in the text explicitly, and that the recorded effects on a given neuron could be due to direct and/or indirect (polysynaptic) mechanisms.

Done – We wrote (p3): “(note that our recordings contain a mixture of directly and indirectly activated neurons)”

Throughout the manuscript we refer to recordings of excitatory neurons and inhibitory- (or inter-) neurons, as opposed to the activated pyramidal and PV populations, respectively.

-It would be helpful to know the quantitative metrics that were used to separate recorded units into putative excitatory and inhibitory, with some reference to previous publications that use those metrics.

Please see Extended Data Fig. 13. Note the additional descriptions in the Data Analysis chapter, with given references (Senzai, Y et al., 2019; de Filippo et al., 2021, previously lines 367-368).

-The authors should clarify whether the recordings were performed in awake or anesthetized animals - the latter is assumed based on the methods, but it's not explicitly stated. Assuming they're anesthetized, the authors should discuss how this could affect the interpretation of their results in the context of other findings that have largely been in awake animals - cortical dynamics are very different under anesthesia, and inhibition is particularly affected.

Done – Now stated in the first paragraph of the result section.

We also added to the discussion: “Finally, the impact of 5-HT signaling may differ under wakeful conditions compared to the anesthetized state during which inhibition and 5-HT levels are affected (Mukaida et al., 2007; Ma et al., 2023). In fact, we demonstrated recently that 5-HT-induced suppressive effects are less pronounced under awake conditions as compared to anesthetized preparations (Azimi et al., 2020).”

-Line 69: “In sharp contrast to the overall activating effect across the pool of inhibitory neurons” should this be “excitatory” neurons?

Sentence rewritten

-Figure 1A: consider labeling pyramidal and PV neurons.

Done

-Figure 1B legend: “without affecting baseline levels of activity.”

Done

-Figure 2A/B: The color scheme becomes confusing here. Labeling the columns as recorded excitatory/inhibitory and rows as activated excitatory/inhibitory (which requires switching the graphs in B), or something similar, could provide more clarity.

Amended – We now assigned colors to the data (traces and bars) of each recorded neuron type, while indicating direct and indirect effects by dark bold and more transparent coloring, respectively. In addition, we labelled each of the columns as “direct effect” and “polysynaptic effect”, respectively.

-Figure 2B: The OI+ and OI- traces that were added to the left panel should also be added to the other panels as well (including in A) or removed.

Done – We now split depiction and analysis, such that we show OI+ and OI- in separate traces and bar plots for direct and indirect effects in both excitatory and inhibitory populations (new Fig. 2c-e and h-j).

-Figure 2D: The OI plot and histogram nicely show the bidirectional change for inh and largely unidirectional for exc, but in as with the above comment the bar plot on the left should only have the OI+ and OI- broken out for both the inh and exc data or should not be done at all. Also, swapping the order of the exc/inh data in the OI plot to match the one above would be more intuitive

Done as suggested – please see reply above.

-Figure 3: The authors should again consider changing the order of the 4x4 plots in A/B as suggested in Figure 2 comments.

Color scheme improved. Moreover, as done in Fig. 2, we labelled each of the columns as “direct effect” and “polysynaptic effect”, respectively.

-Figure 4: It would be useful to see how varying the %units with activated 5-HT2A affect firing rates and changes in magnitude for the “all neurons” condition. For example, a plot with x-axis %excitatory, y-axis %inhibitory, and colormap for firing rate or change in magnitude. This would help the reader understand how the effects of 5-HT2A activation in the two populations interact.

Done - New Extended Data Fig. 11 – Following the reviewer’s recommendation, we generated figures depicting effects on spontaneous activity with varying % of activated excitatory and inhibitory units during systemic activation (Extended Data Fig. 11a). They show how important it is that the balance between excitation and inhibition is maintained in the fluctuation-driven regime. If the perturbation is unbalanced, the network’s baseline activity changes dramatically. The balance between excitation and inhibition is similarly important for the stimulus evoked responses (Extended Data Fig. 11b).

Our results therefore suggest that the balance must be maintained quite precisely, which requires homeostatic mechanisms in the network. It remains to be studied what mechanisms are capable of doing so.

-The authors might consider alternative color schemes rather than red/green to help color-blind readers more easily interpret the figures.

We changed colors throughout all figures, thanks for hinting on this.

Reviewer #3 (Remarks to the Author):

In this manuscript, Barzan and colleagues report the effects of activating the 5-HT_{2A} receptor on visual cortical activity in mice. Receptor activation was achieved via a novel construct that supposedly allowed for optogenetic control of the receptor. They then proceeded to show effect of the manipulation on visually evoked activity.

The study is challenging to understand because key controls are missing. One, there is hardly any calibration and validation of the mOpn4L-5HT_{2A} that allows them to manipulate the receptor signaling. Does the construct work as intended? How do we know it cause 5-HT_{2A}-like function *in vivo*? Where is the mutant protein localized in the cell? None of this was presented quantitatively.

We thank the reviewer for their helpful comments. These points were now integrated in the revised version (see also reply to reviewer #2). Please note that a variant of our chimeric construct has been quantitatively tested with respect to its functionality and localization in 5-HT_{2A} receptor domains including *in vivo* approaches (see ref. Eickelbeck, et al., 2019).

In the introduction we wrote: “Importantly, the C-terminus of the used construct warrants its expression in the endogenous receptor-specific cellular domains (Eickelbeck et al., 2019), providing a light-activatable functional equivalent to endogenous 5 HT_{2A} receptor signals. Moreover, targeting GPCR tools to receptor-specific domains triggers downstream kinetics with similar strength as compared to their endogenous analogues and ensures no overshoot when comparing light-activated to the native cellular effects (Oh et al., 2010).”

New Fig. 1g-i and Extended Data Fig. 1: We now show and quantify directly in V1 brain slices that our construct activates specifically the G_q-coupled pathway.

Second, the characterization of visual responses is very simplistic. Much more can be tested by varying the contrast of the stimuli or the orientation tuning. It is not clear if the manipulation changes contrast response or orientation tuning, because the authors tested only a single grating at a single contrast, which severely limit the evidence supporting their claim for whether the suppression was divisive or subtractive.

As these two points are foundational to understand the study, but are unfortunately missing, it is not obvious that the authors have done what they set out to do.

Varying contrast: We agree with the reviewer that further stimulus variations could further highlight our findings. However, varying contrast would not fundamentally change our conclusion that activation of the 5-HT_{2A} signaling pathway induces divisive gain reduction in populations of V1 neurons. Varying contrast might instead specifically help to investigate properties of contrast normalization in the system.

Please also note that 5-HT-dependent normalization of response magnitude was extensively tested in a previous study of us, where we showed that optogenetically controlled release of 5-HT leads indeed to contrast normalization of response magnitude, most likely involving 5-HT_{2A} signaling, as suggested by pharmacological manipulations in this study (Azimi et al., 2020). However, we agree with the reviewer that we should be explicit regarding these points.

We therefore added (p5-6): “Whether the observed divisive suppression (Fig. 4c and d) displays properties of normalization remains to be tested with different stimulus contrasts. A previous study, in which release of 5-HT was optogenetically controlled, showed normalization of the magnitude of V1 responses to varying contrast, indeed most likely involving 5-HT_{2A} signaling as suggested by additional pharmacological manipulations (Azimi et al., 2020).”

Nonetheless, we performed several new experiments with different stimulus contrast in our settings (see below figure for measurements in PV-Cre mice). These data support the hypothesis that our manipulation induces divisive gain reduction. However, the data were not sufficient to draw specific conclusions about contrast normalization. Thus, we feel that due to the limited sampling number and the fact that this topic was not in the focus of the current study, it would be better to keep these results as preliminary and for the reviewer’s eyes only.

Responses of excitatory single units to varying contrast with and without photoactivation of the 5-HT_{2A} pathway in PV neurons. **a**, Difference between response amplitudes without (control condition) and with photoactivation for different visual stimulus contrasts. Visual responses were normalized to the amplitude of the response to 100% contrast. **b**, Quantification as in Fig. 3f. Comparison between the magnitude of visual responses evoked by stimulus #1 and the average of magnitude values obtained for stimuli #2-4 (including responses to 25, 50, and 100% contrast) for each excitatory unit, during optogenetic activation of 5-HT_{2A} in parvalbumin interneurons. Control (V, black circles), photostimulation conditions (V_{ph}, blue circles). The regression equations are depicted with corresponding colors. Right: regression coefficients (mean \pm s.e.m.). Significant negative value of the coefficient β_4 indicates divisive reduction of the magnitude.

Orientation tuning: Our silicon probes yield simultaneous recordings of multiple cells with different preferred orientations. Thus, using a single orientation stimulates a subset of neurons with an optimal orientation, while others are stimulated at orientations at the flanks of their tuning curves. Hence, the reviewer is right that sampling with different orientations would be desirable, as it would allow us to investigate effects on the shape of tuning curves of individual cells (please note that horizontal and vertical gratings were used in different experiments). However, a larger stimulus set would require longer recording times, which we tried to avoid to account for recording stability during the already long-lasting recording times. These were necessary due to the relatively large inter-stimulus interval needed to reset the optogenetic tool to baseline levels as well as to make sure that neuronal activity returned to baseline before starting a new trial.

To emphasize limitations of interpreting our data as pointed-out by the reviewer we wrote (p6): “Note also that our recordings include neurons with different preferred orientations. Thus, our visual stimulation with a single orientation yields an average population picture of gain suppression rather than scrutinizing effects on individual tuning curves. Hence, possible mixtures of divisive and subtractive or purely subtractive suppression at the single neuron level (Seybold et al, 2015) might be undetected.”

Major comments:

- Control data is missing to show that photo-stimulation of mOpn4L-5-HT_{2A} activates the G-protein signals in the targeted cell types. It is also not clear where this construct is localized in the cell. More importantly, there is no data showing that activation of this construct, actually drives downstream signaling in comparable manner as normal 5-HT_{2A} receptors. Put in other words, what is the difference between the current approach with other methods like Gq-DREADD which also targets the Gq signaling?

Largely addressed above.

We used a light-controlled approach in order to scrutinize the neuronal dynamics following activation of the 5-HT_{2A} receptor pathway at high temporal resolution. The reviewer is right that investigating G_q signaling using DREADD could, in principle, be an alternative. However, besides its lack in millisecond temporal control, such manipulation would additionally involve other (e.g. plastic) changes of the system acting on longer time scales, which was not in the focus of our current study.

- Figure 2b, separate depiction of SphOI+ and SphOI- groups do not seem justified as these are not distinct groups (looking at histogram in Fig. 2d). It's a continuous spectrum of responses. As the author stated in line 62 “a distribution of the OI that is symmetrically centered around 0”. This then carry over to Fig. 2d, where now statistical significance was separately determined for SphOI+ and SphOI- group - of course these are large because the authors arbitrarily divided the group based on the response and then tested the significance of such responses.

We agree with the reviewer and followed also the reviewer #1 request to perform this analysis more in depth and in addition for excitatory cells:

Quantification of activity changes shows a clear increase in activity in neurons with OI-positive (Fig. 2e-g, NEX-Cre, excitatory neurons: +76±13%, inhibitory neurons: +169±59%), while a

more modest decrease in OI-negative neurons (NEX-Cre, excitatory neurons: $-19\pm 3\%$, inhibitory neurons: $-20\pm 3\%$). The same quantification in experiments in PV-Cre mice reveals a strong excitatory effect in interneurons with only a weak increase in activity of excitatory neurons (Fig. 2h-j, inhibitory neurons, OI+: $83\pm 15\%$, excitatory neurons, OI+: $+17\pm 6\%$), and a similar inhibitory effect across both populations (inhibitory neurons, OI-: $-39\pm 4\%$, excitatory neurons, OI-: $-41\pm 2\%$). These values are now stated in the main text (p4).

Please note, that we performed further model analyses and found that when 5-HT_{2A} is activated in the Inh-subpopulation (dark blue in new Extended Data Fig. 8b), the directly stimulated units show increase in OI, while indirectly affected Inh-units show a decrease in OI. In contrast, the Exc-subpopulation, which is only indirectly stimulated, maintains a fairly compact OI distribution. This prediction matches the experimental data (new Fig. 2h) where the shape of the OI distribution of the Exc-population remains roughly similar to the control condition while the distribution of the Inh-population becomes much broader following photostimulation. Analog patterns of widening of the OI distributions emerge when 5-HT_{2A} is activated in the Exc-subpopulation (compare new Extended Data Fig. 8a and Fig. 2c).

Overall, the newly performed analysis supports the effects of 5-HT_{2A} pathway manipulation as previously described: with strong excitation in pyramidal neurons, while activation in PV interneurons causes a mixed effect in inhibitory neurons and mostly an inhibition in excitatory neurons.

- The way the visual responses were mapped is rudimentary. Much more can be tested by varying the contrast of the stimuli or the orientation tuning. It is not clear if the manipulation changes contrast response or orientation tuning. Here the authors show what they say is a divisive suppression, but that's divisive across the population of cells recorded. If they had varied the contrast, then we can see if it is divisive for single neurons, or whether some neurons may show divisive suppression while others have subtractive suppression.

Referred to above, limitations stated in the manuscript.

To improve our study, we performed several complex new experiments. Thereby, due to the given time constraints, we prioritized experiments with respect to what we felt were most related to our main conclusions. In view of the limited time, we feel that additional experiments with varying multiple stimulus parameters, like contrast and orientation, would go beyond the focus of the current study, which shows, for the first time, consequences of a single and specific GPRC pathway activation in distinct cell types *in vivo* together with network modeling. We hope very much that the reviewer can agree with our reasoning.

Minor comments:

- Fig. 1d - what is the fidelity of separation into regular and fast-spiking populations? The authors should provide a histogram so we can see if the distribution of the waveform features used to make the split.

Histogram provided in new Extended Data Fig. 13.

- Fig. 1d - you can say these populations are fast-spiking and regular-spiking cells, but we do not know if these are true excitatory and inhibitory neurons

Amended – We clarify in Methods (p15): “Based on the trough-to-peak time (Extended Data Fig. 13), units were classified as putative inhibitory (< 0.5 ms) or putative excitatory units (> 0.5 ms)^{43,66}.”

In Fig. 1 legend we added: “ ... allowed source separation of responses of putative excitatory or inhibitory neurons based on analysis of waveform features (see Methods).”

- What is the specificity for pyramidal neuron targeting using the NEX-Cre mice? That is, what fraction of transgene-expressing cells are not pyramidal neurons, and what fraction of pyramidal neurons did not express the transgene? This is not quantified.

To quantify specificity, we used a floxed tdTomato construct without C-terminus, as this leads to dominant expression in the cell body, and therefore eases counting of neurons. GluR2/3 was used as a marker for pyramidal cells (please see figure below).

Histological analysis of functional AAV-expression in the V1 region of NEX-Cre mice. a, Quantification of virus-expressing cells in NEX-Cre mice. The AAV1.CAG.FLEX.tdTomato virus (Addgene #28306) was injected into the V1 region. Top: Total number of counted cells per animal (mean + s.e.m.; n = 3 mice); two-tailed paired t-test reveals no significant difference (p = 0.191) between tdTomato positive cells (tdTomato) and tdTomato cells that are also positive for the pyramidal cell marker GluR2/3 (tdTomato / GluR2/3). Bottom: Pie chart reflecting the percentage of tdTomato expressing cells that are also positive for GluR2/3. b, Virus-expressing cells in the V1 region of NEX-Cre mice co-express the native 5-HT_{2A} receptor. The immunohistochemical staining against the native 5-HT_{2A} receptor indicates colocalization between virus-expressing cells (tdTomato) and the receptor (5-HT_{2A}). Rectangles in the left images represent the respective enlarged image on the right. White arrows highlight several co-expressing cells. Scale bars: 150 μm (left); 100 μm (right).

We found that 89.4% of all neurons expressing tdTomato were also GluR2/3 positive. However, please note that this does not mean that the neurons positive for tdTomato but negative GluR2/3 are not pyramidal neurons (Gutiérrez-Igarza et al., 1996; Xu et al., 2003).

Moreover, as the virus was delivered via injection, the number of infected neurons depends on various factors (e.g. virus spread and distance to injection site). Therefore, to quantify the fraction of all pyramidal cells expressing the construct would be difficult and prone to error. We would thus prefer to not state a particular staining value here.

References:

Gutiérrez-Igarza, K., Fogarty, D., Pérez-Cerdá, F., Doñate-Oliver, F., Albus, K., & Matute, C. (1996). Localization of AMPA-selective glutamate receptor subunits in the adult cat visual cortex. *Visual Neuroscience*, 13(1), 61-72. doi:10.1017/S0952523800007136.

Xu, L., Tanigawa, H. and Fujita, I. (2003), Distribution of α -amino-3-hydroxy-5-methyl-4-isoxazolepropionate-type glutamate receptor subunits (GluR2/3) along the ventral visual pathway in the monkey. *J. Comp. Neurol.*, 456: 396-407. <https://doi.org/10.1002/cne.10538>.

- Line 49 - provide more specifics about the optogenetic activation (pulse duration, frequency, etc.)

Clarified – We added (p13): “In photostimulation conditions, continuous light (465 nm, applied with LED driver and module system, Plexon) lasted 8 s, which in the V_{ph} condition started 1.8 s after the 1st visual stimulus onset.”

- Figure 2c and d, left panels, what are those colored bars at the top of the plots?

After rearranging the figure and changing the color scheme, the bars (indicating the paradigm) were removed.

- Figure 2c and d, the right panel’s statistical test was specified as paired t-test. What about the left and middle panels?

Information added – The paired test was used for Fig. 2c, d left and middle panels. Kolmogorov-Smirnov test was used for comparing distributions. This is now stated accordingly in the legend.

- It’s recommended to avoid using red and green as the colors because they are not easily distinguished for people who are colorblind.

Done – We changed colors throughout all figures, thanks for hinting on this.

- The description of 'polysynaptically transmitted hyperpolarization via PV neurons in which the 5-HT_{2A} receptor was photoactivated.' was confusing and it is not clear what the authors mean (line 74/75)

Thanks for the remark, now corrected (p4): “Thus, the observed dominant decrease in excitatory cells’ spontaneous activity results most likely from hyperpolarization transmitted via PV neurons in which the 5 HT_{2A} receptor pathway is photoactivated.”

- There is some labeling but it is thought that much of the endogenous 5-HT_{2A} receptors are localized to apical dendrites.

Indeed our 5-HT_{2A} construct was mostly localized on the apical dendrites, as can be observed in Extended Fig 2c and e, where apical dendrites cross from layer 5 towards layer 2/3. Clear somatic expression as in the PV interneurons (Extended Data Fig 3b top) was rarely visible.

In PV interneurons the localization of 5-HT_{2A} is indeed most likely on both soma and dendrites, as expected from the literature (Weber and Andrade, 2010), the below attached figure shows an example. Interestingly, labeling may depend on the used antibody (Weber and Andrade, 2010).

From left to right: Immunohistochemical staining against 5-HT_{2A} (green), tdTomato (red) and merged image.

- It is not appropriate to use a single orientation to measure the visual responses, as V1 neurons exhibit different tuning so single-orientation will drive few cells well, but also many cells suboptimally, affecting how one may consider the 'average response'.

Please see our reply above.

- Fig. 3 includes result of pyramidal neuron stimulation, but the title only mentioned stimulation of parvalbumin interneurons

Rewritten.

Reviewers' comments:

Reviewer #1 (Remarks to the Author):

The authors have addressed most of my concerns. Especially, the new simulations in the mean-driven regime are quite useful to distinguish the emergence of the reported effects in fluctuation-driven regimes.

Here are some minor comments:

L.40-44: The opening sentence of the Results is long and difficult to parse

L. 73: “overall slight but not significant ..”

Rewrite

L. 85-86: “... the effect following [...] was overall activity across ..”

Rewrite

L. 146-147: “... yields an average population picture of gain suppression ..”

Rewrite

L. 190: “Hence, for the mean-driven the outcome was dramatically different from the experimental observations”

—> change to: Hence, we observed different results for the mean-driven regime (or something similar)

L. 288: “ Thus, the detailed picture of the underlying mechanisms in terms of layer-specific circuitries needs further study.“

Rewrite

L. 289: I don't understand what this is exactly referring to:

“expression strength of optogenetic tools can vary“

In general, the writing can still be improved - there are examples of long and complicated sentences that can be improved for better readability.

One example is L.297-300:

“Our major point that the joint activation of the 5-HT_{2A} pathway (i.e. simultaneous Gq-signaling in pyramidal and PV neurons) in a balanced regime leads to a robust qualitative change in the cortical state allowing for gain control of external input without changes in baseline activity is further supported by previous data in the native system.“

Reviewer #2 (Remarks to the Author):

Barzan et al. have performed a significant amount of work (experiments, analysis, and text revisions) to substantially improve this manuscript and resolve previous issues - in particular, clarification of the optogenetic tool employed and directly linking experimental and modeling data. The combination of physiological and computational techniques to address the mechanisms of 5-HT_{2A} pathway modulation of cortical network function is of broad interest to the neuroscience community. We thank the authors for a clear and organized response to reviewer comments, and applaud their efforts to enhance this story.

Minor comments:

-It's not entirely clear what the meaning of “ongoing brain broadcasts” is in the abstract. Perhaps a different phrase (e.g., “ongoing neural dynamics”) that more clearly describes what the authors address would be appropriate?

-Line 47, “respectively” might be expected to imply individual reference to the NEX and PV mice, (e.g., excitatory and inhibitory neurons), however there is not such a statement in the sentence. This could likely be omitted unless I'm missing something.

-The paragraph on line 141 could be moved to the Discussion.

Reviewer #3 (Remarks to the Author):

Our initial round of review has critical comments echoing those raised by Reviewer 2. Namely, the approach does not alter endogenous 5-HT2A receptor signaling, but rather activates some signaling with unclear consequences because it is a chimeric construct with light-activated melanopsin targeted into 5-HT2A receptor domains. It is an artificial protein with different components fused. The authors themselves acknowledge that “our construct does not concur with the normal complex distribution of 5-HT2A receptors across cortical layers”, in that it has obvious differences with endogenous 5-HT2A receptors.

Therefore, what this paper shows is that manipulation of the cortical circuits with this specific construct have certain influence on neural activity dynamics. While that itself could be of theoretical interest, that would require us knowing more about how this construct act at molecular and cellular levels, and those details are missing. The authors noted that a variant of this construct has been tested, but not the specific construct that they are studying here. They cited a reference to say “targeting GPCR tools to receptor-specific domains triggers downstream kinetics with similar strength” but does not provide data to support such statement for their construct. Instead of performing standard assays such as BRET or calcium mobilization to investigate Gq and other signaling cascades, they rely on calcium imaging in brain slices where the observed fluorescence changes are several steps away from potential Gq signaling and not easy to interpret.

The results do not provide insights into how endogenous 5-HT2A receptors work, given the obvious difference between their engineered construct and endogenous receptors. This was a major comment that is foundational to understanding the results, but the authors did not do the work needed to change that view.

The second major comment was regarding the lack of visual characterizations like contrast tuning and orientation tuning that are common in visual neurophysiology. The authors chose to disregard these as limitations in the discussion section. The issue is that these characterizations are needed for constraining computational models – for example, see any sensory cortical circuit modeling study by Scanziani, Carandini/Harris, and Geffen labs. There are too many synaptic connectivity and strengths parameters and often models are overfit if the basic characterizations are not performed.

Authors' reply to the reviewer comments

We thank all reviewers for their helpful comments on our revised manuscript. We are glad that reviewer #1 found our new simulations useful and that reviewer #2 appreciated the clarification of our optogenetic tool. We have carefully addressed the remaining minor comments of both reviewers in our current version (please see details below). However, reviewer #3 requested more information about how our tool acts at molecular and cellular levels in comparison to endogenous 5-HT_{2A} receptors and further requests data related to visual stimulation. To address these concerns, we now performed a number of GsX-assays showing the similarity in molecular downstream signaling between the endogenous 5-HT_{2A} receptor pathways and of our construct. In addition, we expanded our Ca²⁺-imaging dataset and now show the close similarity in the PLC-dependent Ca²⁺-signal comparing light activation of the designed Opm4L-5-HT_{2A}CT chimeric construct and 5-HT-activation of 5HT_{2A}-receptors. Furthermore, we added electrophysiological data showing the effects of our tool upon optogenetic stimulation on visual responses to various stimulus contrasts and orientations.

Point-by-point responses to the individual reviewer comments:

Reviewer #1 (Remarks to the Author):

The authors have addressed most of my concerns. Especially, the new simulations in the mean-driven regime are quite useful to distinguish the emergence of the reported effects in fluctuation-driven regimes.

We thank the reviewer for all of their helpful comments and for taking the time to review our manuscript.

Here are some minor comments:

L.40-44: The opening sentence of the Results is long and difficult to parse

Sentence shortened.

L. 73: "overall slight but not significant .." Rewrite

We removed "overall".

L. 85-86: "... the effect following [...] was overall activity across .." Rewrite

We removed "overall".

L. 146-147: "... yields an average population picture of gain suppression .." Rewrite

Sentence rewritten (and moved to page 9, bottom).

L. 190: "Hence, for the mean-driven the outcome was dramatically different from the experimental observations" —> change to: Hence, we observed different results for the mean-driven regime (or something similar)

Done.

L. 288: “ Thus, the detailed picture of the underlying mechanisms in terms of layer-specific circuitries needs further study.” Rewrite

Rewritten.

L. 289: I don’t understand what this is exactly referring to: “expression strength of optogenetic tools can vary“

Rewritten.

In general, the writing can still be improved - there are examples of long and complicated sentences that can be improved for better readability.

One example is L.297-300: “Our major point that the joint activation of the 5-HT2A pathway (i.e. simultaneous Gq-signaling in pyramidal and PV neurons) in a balanced regime leads to a robust qualitative change in the cortical state allowing for gain control of external input without changes in baseline activity is further supported by previous data in the native system.“

Sentence reformulated.

We carefully checked again the writing together with another native speaker.

Reviewer #2 (Remarks to the Author):

Barzan et al. have performed a significant amount of work (experiments, analysis, and text revisions) to substantially improve this manuscript and resolve previous issues - in particular, clarification of the optogenetic tool employed and directly linking experimental and modeling data. The combination of physiological and computational techniques to address the mechanisms of 5-HT2A pathway modulation of cortical network function is of broad interest to the neuroscience community. We thank the authors for a clear and organized response to reviewer comments, and applaud their efforts to enhance this story.

We thank the reviewer very much for their encouraging pleasant and helpful comments.

Minor comments:

-It’s not entirely clear what the meaning of “ongoing brain broadcasts” is in the abstract. Perhaps a different phrase (e.g., “ongoing neural dynamics”) that more clearly describes what the authors address would be appropriate?

For clarification changed to “neuronal dynamics”.

-Line 47, “respectively” might be expected to imply individual reference to the NEX and PV mice, (e.g., excitatory and inhibitory neurons), however there is not such a statement in the sentence. This could likely be omitted unless I’m missing something.

We removed “respectively”.

-The paragraph on line 141 could be moved to the Discussion.

Done.

Reviewer #3 (Remarks to the Author):

Our initial round of review has critical comments echoing those raised by Reviewer 2. Namely, the approach does not alter endogenous 5-HT_{2A} receptor signaling, but rather activates some signaling with unclear consequences because it is a chimeric construct with light-activated melanopsin targeted into 5-HT_{2A} receptor domains.

We thank the reviewer again for their critical comments. The reviewer is correct in that their previous critics were largely echoing those of reviewer #2. In fact, reviewer #2 now states that our revised manuscript “resolve previous issues - in particular, clarification of the optogenetic tool employed and directly linking experimental and modeling data”. In our previously revised version we used an antibody approach (applying PLC antagonist) to verify that our tool activates the G_q pathway similar as known for endogenous 5-HT_{2A} signaling. However, we agree with the reviewer that testing of further potential intracellular downstream targets is important. We have therefore performed additional experiments as suggested by the reviewer using established GsX-assays (Ballister et al., BMC Biol 2018, Zhou et al. Nat Commun 2023) and Ca²⁺-imaging, and compared light-activation of the designed Opn4L-5HT_{2A}CT chimeric construct and drug-activation of 5-HT_{2A}-receptors (new Extended Data Fig. 2). These new data revealed highly similar G protein selectivity and Ca²⁺-mobilization patterns, suggesting similar intracellular downstream signaling of the endogenous receptor and of our tool.

Extended Data Fig 2. GsX assay and G_q-dependent calcium imaging. **a/b**, GsX assay reaction presented as mean luminescence (colored traces) and s.e.m. (shaded areas) for Opn4L-5-HT_{2A}CT (stimulated with a 5 s blue light pulse in **a**) or 5-HT_{2A} receptor (stimulated with 20 μM 5-HT agonist Ro 60-175 Fumarate in **b**), co-transfected with either the GloSensor only or the G_s chimera for G_i, G_o, G_q, and G_s (n=4 wells for each chimera). Vertical lines and arrows indicate stimulation onset. **c**, Luminescence values (normalized) after stimulation onset indicate a significant preference for G_q pathway activation over G_i, G_o and G_s for both Opn4L-5-HT_{2A}CT and the 5-HT_{2A} receptor (n=30 post stimulation values for each condition; *p<0.05, one-way ANOVA followed by all pairwise multiple comparison procedures (Tukey Test)). **d**, 2-Photon calcium imaging shows increased fluorescence after

stimulation of Opn4L-5HT_{2A}CT with light (965 nm laser, dashed blue line) or 5-HT_{2A} receptor activation with 50 μM 5-HT (green shaded area). Blocking the G_q actuator PLC with 10 μM U73122 abolishes the calcium increase completely. Opn4L-5HT_{2A}CT: n=3 dishes, 103 cells; Opn4L-5HT_{2A}CT + antagonist: n=3 dishes, 130 cells; 5-HT_{2A} receptor: n=3 dishes; 60 cells; 5-HT_{2A}CT receptor + antagonist: n=3 dishes, 68 cells.

It is an artificial protein with different components fused. The authors themselves acknowledge that “our construct does not concur with the normal complex distribution of 5-HT_{2A} receptors across cortical layers”, *in that it has obvious differences with endogenous 5-HT_{2A} receptors*.

It is a common principle of optogenetic tools to be constructed out of different functional components. Therefore, in this respect our construct follows such established principles (see Marcus & Bruchas, Pharmacol. Rev. 2023 for a recent review).

The reviewer is right, and as we have stated, the expression of the construct can vary across cortical layers. However, we show that its activation leads to the same suppressive divisive effects of visual responses as compared to endogenous 5-HT_{2A} receptors (please see the cited work by Michael et al., 2019).

To further clarify this point we added in the main text: “Using subcutaneous injections of 2,5-dimethoxy-4-iodoamphetamine (DOI) to stimulate endogenous 5-HT_{2A} receptors in mice, Michael and colleagues³¹ showed that suppressive effects were more pronounced in layer 2/3 and also cell type specific.”

Therefore, what this paper shows is that manipulation of the cortical circuits with this specific construct have certain influence on neural activity dynamics.

Here we disagree with the reviewer. – See above point, please note that we have reproduced previous *in vivo* data derived in V1, where endogenous signaling was triggered by a 5-HT_{2A} agonist (Michael et al., 2019). In particular, these authors and our study show that while the gain of visual responses is reduced, spontaneous activity levels are unchanged (cf. Fig. 3 in Michael et al., 2019 and cf. Fig. 4 in our revised manuscript). In addition, Michael et al., 2019 showed that suppressive effects were more pronounced in layer 2/3 where the majority of cells shown in Fig. 4 and Extended Data Fig. 10 were recorded (we now added mean recording depth in Fig. 4 legend). Thus, with our tool we replicated specific signatures of neuronal activity and of visual responses when endogenous 5-HT_{2A} receptors were stimulated (see also Azimi et al. 2020).

Given the confirmation of recent results of the literature we are confident that the endogenous 5-HT_{2A} effects on neuronal activity dynamics and the effects of our tool are highly similar.

Importantly, in our current manuscript we go several significant steps further: i) we isolate neuronal activity based on cell type-specific 5-HT_{2A} pathway activation. This means we are able to provide ii) experimental predictions with respect to systemic 5-HT_{2A} pathway activation (gain suppression, stable baseline), and based on the revealed data iii) we summarize the underlying mechanisms by a model.

While that itself could be of theoretical interest, that would require us knowing more about how this construct act at molecular and cellular levels, and those details are missing. The authors noted that a variant of this construct has been tested, but not the specific construct that they are studying here. They cited a reference to say “targeting GPCR tools to receptor-specific

domains triggers downstream kinetics with similar strength” but does not provide data to support such statement for their construct.

Addressed, please see above. We have added new data providing further details of how the construct acts on downstream targets at cellular level using GsX-assays and Ca²⁺-imaging.

Note that the molecular and the cellular activation pattern of a functional identic construct has been shown previously by Eickelbeck et al. (2019). The only difference between the construct used and the described construct is that the published one has an additional GCaMP6m for visualization of Ca²⁺-signals fused to the C-terminus. As shown in different neuronal populations *in vitro* and *in vivo* by Eickelbeck et al. (2019) the specific localization of the GPCR to its receptor domain (and not the addition of the Ca²⁺-sensor) is mediated by the C-terminus of the 5-HT_{2A} receptor altering its distribution and activation dynamics to similar intracellular Ca²⁺-signals (in contrast to the untagged construct) as observed for endogenous 5-HT_{2A} receptors. Similar results for receptor-specific targeting mediated by the C-terminus have also been demonstrated for the CT of 5-HT_{1A} (Oh et al., 2010, Masseck et al., 2014), 5-HT_{1B} (Hasegawa et al., 2017), and 5-HT_{2C} (Spoida et al. 2014) when fused to rod and cone opsins and melanopsin.

Instead of performing standard assays such as BRET or calcium mobilization to investigate Gq and other signaling cascades, they rely on calcium imaging in brain slices where the observed fluorescence changes are several steps away from potential Gq signaling and not easy to interpret.

Addressed above. We now performed the standard GsX-assay as a bioluminescent readout assay for interrogating G protein selectivity.

The measured Ca²⁺-signals show a direct dependency on intracellular PLC activation (the major downstream target of G_q signaling). Therefore, the statement that these data are “not easy to interpret” appears somewhat exaggerated.

We now additionally measured the Ca²⁺-signals in HEK cells and show similar PLC-dependent Ca²⁺-signals in response to 5-HT-activation of 5-HT_{2A} receptors and activation by our Opn4L-5-HT_{2A}CT construct (new Extended Data Fig. 2d).

The results do not provide insights into how endogenous 5-HT_{2A} receptors work, given the obvious difference between their engineered construct and endogenous receptors.

Given the similarities in G_q pathway activation, expression in 5-HT_{2A}-specific domains, in intracellular downstream signaling (see new GsX-assays, new Extended Data Fig. 2a-c), and in neural dynamics and visual responses (as confirmed by systemic activation and comparison to previously published literature), we think that our tool is suitable to mimic 5-HT_{2A} pathway activation, specifically its impact on neuronal activity dynamics.

This was a major comment that is foundational to understanding the results, but the authors did not do the work needed to change that view.

We thank the reviewer for their continued criticism and motivation to further analyze potential differences in subcellular signaling between the endogenous 5-HT_{2A} pathway and of our construct. As suggested by the reviewer we have therefore performed standard assays and added those data to the manuscript.

The second major comment was regarding the lack of visual characterizations like contrast tuning and orientation tuning that are common in visual neurophysiology. The authors chose to disregard these as limitations in the discussion section.

Possible misunderstanding – In our previous manuscript we did not add further visual data as in our opinion they are beyond the scope of our study (summarized above, i-iii).

We agree that data based on stimulus parameter variations are principally helpful. We added visual data that additionally present visual responses to various orientations and contrasts following photostimulation of the 5-HT_{2A} receptor pathway in PV neurons (new Extended Data Fig. 8 and new Extended Data Fig. 9).

Extended Data Fig. 8. No change for the preferred grating orientation following photostimulation of the 5-HT_{2A} receptor pathway in PV neurons. **a**, Examples of single cell responses to grating stimuli of various orientations (including opposite directions) presented at 100% contrast, in the absence (V) and presence (V_{ph}) of photostimulation, black and blue lines, respectively. Responses were normalized to maximum response over all orientations and both conditions (averages across 7-10 stimulus repetitions). **b**, Preferred orientation of the neurons shown in **a** without (“light off”) and with (light on”) photostimulation. Preferred orientation is only minimally affected by photostimulation. Light red markers indicate excitatory units and blue marker indicates inhibitory unit.

Extended Data Fig. 9. Responses of excitatory single units to varying contrast with and without photostimulation of the 5-HT_{2A} pathway in PV neurons. Quantification as in Fig. 3f. Comparison between the magnitude of visual responses evoked by stimulus #1 and the average of magnitude values obtained for stimuli #2-4 (including responses to 25, 50, and 100% contrast) for each excitatory unit, during optogenetic activation of the 5-HT_{2A} pathway in parvalbumin interneurons. Control (V , black circles), photostimulation conditions (V_{ph} , blue circles). The regression equations are depicted with corresponding colors. Right: regression coefficients (mean \pm s.e.m.). Significant negative value of the coefficient β_4 indicates divisive reduction of the magnitude.

As we added a new experiment where orientation preference was measured, please note that Michael and colleagues (2019) have indeed shown that endogenous 5-HT_{2A} signaling does not alter orientation tuning. We further added in the main text: “Interestingly, similar to our observations, Michael and colleagues³¹ found no changes in preferred orientation following DOI administration.”

The main focus of our manuscript is to reveal the underlying mechanism of reduction in response gain, to provide modeling of these mechanisms, and to resolve the paradox of how 5-HT_{2A} signaling (although principally activating) leads to suppression of neuronal gain – rather than an extensive characterization of visual processing. Thus, the strength and the novel conclusions that can be drawn from our study lies in the elucidation of a mechanism which involves 5-HT_{2A}-induced modulation of gain reduction that simultaneously maintains baseline levels of ongoing activity.

To achieve these goals our study performed novel *in vivo* optogenetic control of GPCR-signaling in different types of cortical neurons in awake animals.

The issue is that these characterizations are needed for constraining computational models – for example, see any sensory cortical circuit modeling study by Scanziani, Carandini/Harris, and Geffen labs. There are too many synaptic connectivity and strengths parameters and often models are overfit if the basic characterizations are not performed.

We agree that overfitting can be a serious issue in computational modeling. To avoid overfitting in our model, we employed a spiking cortical network developed by Sadeh et al. (2017) with as few parameters as possible to characterize our experimental observations. As we were interested in modeling the mechanisms that control the gain of external inputs by simultaneously stabilizing baseline levels, rather than characterizing processing certain visual input by a neural circuit, we kept the model as simple as possible, downscaling it to the major experimental constraints. Specifically, we do not fit individual synaptic connections (such as machine learning methods would), but only fix the probability of a connection and the means and variances of weights across entire populations. In other words, whether a certain synaptic connection exists and its weight are drawn randomly for each instance of the network in our simulations. The outputs of the model shown in this manuscript are always averaged across multiple different network instantiations, thus ensuring that the results are robust.

References

- Ballister, E. R., Rodgers, J., Martial, F. & Lucas, R. J. A live cell assay of GPCR coupling allows identification of optogenetic tools for controlling Go and Gi signaling. **BMC Biol.** **16**, 10 (2018).
- Eickelbeck, D. *et al.* CaMello-XR enables visualization and optogenetic control of Gq/11 signals and receptor trafficking in GPCR-specific domains. **Commun Biol** **2**, 1–16 (2019).
- Hasegawa, E. *et al.* Serotonin neurons in the dorsal raphe mediate the anticataplectic action of orexin neurons by reducing amygdala activity. *Proc. Natl. Acad. Sci. U.S.A.* **114**, (2017).
- Marcus, D. J. & Bruchas, M. R. Optical Approaches for Investigating Neuromodulation and G Protein–Coupled Receptor Signaling. **Pharmacol. Rev.** **75**, 1119–1139 (2023).
- Masseck, O. A. *et al.* Vertebrate cone opsins enable sustained and highly sensitive rapid control of Gi/o signaling in anxiety circuitry. *Neuron* **81**, 1263–1273 (2014).
- Michael, A. M., Parker, P. R. L. & Niell, C. M. A Hallucinogenic Serotonin-2A Receptor Agonist Reduces Visual Response Gain and Alters Temporal Dynamics in Mouse V1. *Cell Rep.* **26**, 3475-3483.e4 (2019).

Oh, E., Maejima, T., Liu, C., Deneris, E. & Herlitze, S. Substitution of 5-HT_{1A} Receptor Signaling by a Light-activated G Protein-coupled Receptor. **J. Biol. Chem.** **285**, 30825–30836 (2010).

Sadeh, S., Silver, R. A., Mrcic-Flogel, T. D. & Muir, D. R. Assessing the Role of Inhibition in Stabilizing Neocortical Networks Requires Large-Scale Perturbation of the Inhibitory Population. *J. Neurosci.* **37**, 12050–12067 (2017).

Spoida, K., Masseck, O. A., Deneris, E. S. & Herlitze, S. Gq/5-HT_{2c} receptor signals activate a local GABAergic inhibitory feedback circuit to modulate serotonergic firing and anxiety in mice. *Proc. Natl. Acad. Sci. U.S.A.* **111**, 6479–6484 (2014).

Zhou, F. *et al.* Optimized design and in vivo application of optogenetically functionalized Drosophila dopamine receptors. **Nat. Commun.** **14**, 8434 (2023).

REVIEWERS' COMMENTS

Reviewer #2 (Remarks to the Author):

Barzan et al. have addressed all remaining concerns with this manuscript. The further clarification on the effects of Opn4L-5-HT2ACT activation relative to endogenous 5-HT2A activation, as well as data related to effects on visual orientation tuning, are useful to the reader. Overall, this study provides evidence for a polysynaptic mechanism of gain control in cortex via 5-HT2A activation. Limitations of the study are sufficiently discussed.

Extended Data Fig2: It would be helpful to see the plots in a and b with similar y-axis scaling (e.g., change the max of b to 1200 to match a). The different axis limits currently obscure overall differences in magnitude between effects of the engineered and endogenous receptors. The normalized data in c already provide a useful comparison of the relative activation of the different pathways.

Point-by-point responses to the reviewer comments:

Reviewer #2 (Remarks to the Author):

Barzan et al. have addressed all remaining concerns with this manuscript. The further clarification on the effects of Opn4L-5-HT2ACT activation relative to endogenous 5-HT2A activation, as well as data related to effects on visual orientation tuning, are useful to the reader. Overall, this study provides evidence for a polysynaptic mechanism of gain control in cortex via 5-HT2A activation. Limitations of the study are sufficiently discussed.

Extended Data Fig2: It would be helpful to see the plots in a and b with similar y-axis scaling (e.g., change the max of b to 1200 to match a). The different axis limits currently obscure overall differences in magnitude between effects of the engineered and endogenous receptors. The normalized data in c already provide a useful comparison of the relative activation of the different pathways.

We followed the reviewer's suggestion and scaled the y-axes in Extended Data Fig. 2a and b to 1200 so that they are matched.